# Wnt signaling restores evolutionary loss of robust foot regeneration rates in *Hydra*

Sergio E. Campos [1,2], Sahar Naziri[1,3], Jackson Crane[1], Jennifer Tsverov[1], Ben D. Cox[1], Craig Ciampa[1], Ruthie B. Spencer[4], Robert E. Steele [5], Daniel E. Martínez[4] & Celina E. Juliano [1] ✉

The regenerative potential of animals varies widely, even among closely related species. In a comparative study of regeneration across the *Hydra* genus, we found that while most species exhibit robust whole-body regeneration, *Hydra oligactis* and other members of the Oligactis clade display substantially reduced rates of foot regeneration. To investigate the mechanisms underlying this deficiency, we analyzed transcriptional responses during head and foot regeneration in *H. oligactis*. Our analysis revealed that the general injury response in *H. oligactis* lacks strong activation of Wnt signaling, a pathway essential for *Hydra vulgaris* foot regeneration. Notably, transient treatment with a Wnt agonist in *H. oligactis* triggered a foot-specific transcriptional program, significantly increasing rates of foot regeneration. Our transcriptional profiling also revealed *dlx2* as a likely high-level regulator of foot regeneration, dependent on Wnt signaling activation. Our study establishes a comparative framework for understanding the molecular basis of regeneration in *Hydra* and provides a new platform for investigating the evolution of regenerative mechanisms.

Regeneration is defined as the restoration of lost or injured tissues, appendages or organs[1]. While most animals have some capacity to regenerate, regenerative potential varies widely across animals, from the ability to heal small wounds to the ability to robustly and perfectly rebuild any lost body part. Thus, a fundamental question is: how are some animals capable of regenerating large portions of their body after injury, while others exhibit only a fraction of this capacity? Mapping animal regenerative abilities on the phylogenetic tree suggests a complex evolutionary history, with numerous gains and losses and a wide variation in the robustness of regenerative processes across species[2]. However, the mechanisms driving evolutionary changes in regenerative potential across animals are not well understood for most clades. Notably, regeneration defects or loss of regenerative potential have been reported in several species of planarians, a group known for its high regenerative ability[3]. In planarians, overactivation of the canonical Wnt signaling pathway results in head regeneration defects in some species, consistent with its role in posterior patterning[4–7]. Wnt signaling plays a critical role in regeneration across animals, not only in axial patterning, but also as a part of the general injury response[8–13]. Therefore, modulations in this pathway could be a key driver of evolutionary changes in regeneration potential across animals. To explore this, we require additional comparative studies in clades that contain closely related species with varying regenerative outcomes, which could reveal both commonalities and differences in the mechanisms underlying losses and gains of regenerative capacity in animals. Such studies are a promising avenue to both reveal how regenerative abilities have been shaped throughout evolution, as well as uncover basic mechanisms of regeneration.

Cnidarians, a group that includes sea anemones, jellyfish, and corals, are highly regenerative. Furthermore, given their phylogenetic

[1]Department of Molecular and Cellular Biology, University of California, Davis, CA, USA. [2]Centro de Investigación sobre el Envejecimiento, Centro de Investigación y de Estudios Avanzados del Instituto Politécnico Nacional (CIE-Cinvestav), Sede Sur, Mexico City, Mexico. [3]Department of Neuroscience and Developmental Biology, Faculty of Life Sciences, University of Vienna, Vienna, Austria. [4]Department of Biology, Pomona College, Claremont, CA, USA. [5]Department of Biological Chemistry, University of California, Irvine, CA, USA. ✉e-mail: cejuliano@ucdavis.edu

relationship as sister group to bilaterians[14], research in cnidarians is critical to understand the evolutionary history of regeneration. The cnidarian polyp *Hydra vulgaris* is capable of whole-body regeneration and is a well-established research organism for studying the molecular mechanisms of regeneration. The body plan of animals in the *Hydra* genus is arranged around a single body axis, with a head, composed of a tentacle ring around a hypostome or mouth at the oral end; and the peduncle and an adhesive basal disk (foot) at the aboral end. These structures are connected by a cylindrical body column. When bisected, both halves of *H. vulgaris* are capable of regeneration: the top half robustly regenerates a foot, and the bottom half robustly regenerates a head. Conversely, a few studies have reported that the closely related species *Hydra oligactis* displays markedly reduced rates of foot regeneration[15,16], suggesting regenerative outcomes vary in the genus. However, the evolutionary history and molecular mechanisms underlying foot regeneration defects in the *Hydra* genus remain underexplored.

In *H. vulgaris*, the maintenance of tissue patterning in the uninjured animals relies on a Wnt signaling center located at the oral end, which is necessary and sufficient to pattern the head[8,9]. Upon bisection perpendicular to the oral-aboral axis in *H. vulgaris*, the process of regeneration includes the formation of a new Wnt organizer on the oral-facing wound, which will direct the regeneration of a new head[9,17,18]. However, injury-induced transcriptional activation of Wnt ligands occurs at both sides of the wound as part of the general injury response. Treatment with a Wnt inhibitor impairs or delays both head and foot regeneration, showing that the transient activation of Wnt signaling in foot regeneration has a critical function that remains unknown[18–20]. In planarians, a similar regeneration pattern is observed after bisection, with Wnt activation required at both sides of the wound during the early injury response, followed by restriction to tail regeneration, where it directs posterior development[8,21]. However, in acoels, early Wnt activation after bisection is restricted to aboral wound sites, where it similarly directs posterior development[12], exemplifying variation in the role of Wnt signaling during whole body regeneration. In vertebrates, Wnt signaling is rapidly upregulated early in regeneration, and Wnt inhibition results in impaired regeneration of mouse digits, demonstrating its importance even in animals with more limited regeneration abilities[11,22].

In this study, we confirmed that *H. oligactis* has reduced foot regeneration rates and then surveyed the genus, revealing a likely evolutionary loss of robust foot regeneration outcomes in the Oligactis clade (Fig. 1a). To reveal the mechanisms underlying the reduced foot regeneration rates in *H. oligactis* and to better understand foot regeneration in *H. vulgaris*, we performed RNA-seq over a time course of *H. oligactis* regeneration and compared the results to existing data sets from *H. vulgaris*. Our analysis revealed that injury-induced Wnt signaling activation is significantly attenuated in *H. oligactis*, resulting in slower head regeneration and a foot regeneration block in a large percentage of animals. Using pharmacological manipulation, we demonstrated that activation of the Wnt pathway leads to significantly higher rates of *H. oligactis* foot regeneration. This approach also allowed us to identify key Wnt signaling-dependent transcriptional regulators involved in foot formation, providing new insights into the regulatory control of foot regeneration in *Hydra*.

## Results

### Foot regeneration rates vary across the *Hydra* genus

The common laboratory *Hydra* species, *H. vulgaris*, can robustly regenerate a complete head or foot from body column tissue. By contrast, it was previously reported that *H. oligactis* displays low rates of foot regeneration[15]. To investigate this further, we first characterized the foot regeneration rates in our *H. oligactis* laboratory strain (Innsbruck 12) for which we recently assembled a high-quality genome[23]. We bisected *H. oligactis* midway between the head and

foot (50% bisection), perpendicular to the oral-aboral axis, and monitored the rate of foot regeneration for the top half (Fig. 1b, c). We found that after 6 days, only ~10% of the animals regenerated their feet (Fig. 1b), while the remaining animals remained footless, characterized by a flat aboral end (Fig. 1c) and an inability to stick to surfaces. In our initial experiments, footless animals where continuously monitored for up to 2 months, during which foot regeneration did not occur, indicating that the animals remained stably footless. We did not observe new foot regeneration past 120 h and therefore conservatively set 144 h as the endpoint of our assays moving forward. We also evaluated the absence or presence of a foot using a standard colorimetric assay developed for *H. vulgaris* that reveals the activity of a foot-specific peroxidase at seven days post bisection[24]. We confirmed that morphologically footless *H. oligactis* also lacked foot peroxidase staining (Fig. 1d, e). In contrast to *H. oligactis*, we confirmed that in the same conditions, nearly 100% of *H. vulgaris* regenerate their feet after bisection, consistent with the published literature (Fig. 1h).

We next asked if the rate of successful foot regeneration in *H. oligactis* depends on the location of the cut along the oral-aboral axis. To test this, we performed bisections at three different positions along the body column. In addition to mid-body bisection (50% of body length), we also bisected at 25% body length from the head, just below the tentacle ring, and at 75% body length from the head, just above the peduncle and foot (Fig. 1f). We performed the same bisections in *H. vulgaris* as a positive control for foot regeneration. Consistent with previous literature, *H. vulgaris* exhibited nearly 100% successful foot regeneration after all three amputations, but with different kinetics depending on the level of the amputation (Fig. 1h)[15,25–27]. Also consistent with previous literature, the success rate of *H. oligactis* foot regeneration depended on the location of the cut; after 75% bisection, ~65% of animals successfully regenerated their feet, while foot regeneration did not occur in any animal after 25% bisection (Fig. 1g)[15]. Hoffmeister et al.[24] reported that the foot regeneration rate in *H. oligactis* reached ~30% at 96 h after a 50% bisection, a substantial difference compared to the 10% rate we observed for the same injury. Without access to the previously tested strain, it is not possible to determine whether this difference reflects genetic variability or differences in culture conditions. Notably, we also conducted regeneration assays in the *H. oligactis* Swiss strain and observed regeneration rates closely matching the Innsbruck strain, which could point to the importance of testing regenerative ability in different strains and species across the same culture conditions (Supplementary Fig. 1).

We next tested foot regenerative rates across the *Hydra* genus to better understand the evolutionary history of this trait. While a handful of regeneration studies have been done in different *Hydra* species[15,16,28,29], no systematic study of foot regeneration has been done across the genus in similar culture conditions on validated species. We therefore performed foot amputation assays (Fig. 1f) on three additional *Hydra* species to cover each phylogenetic clade (Fig. 1a, i–k)[30]. Our experiments revealed that both *H. oligactis* and *H. oxycnida*, which are in the Oligactis clade, show similar rates of failed foot regeneration (Fig. 1g, i, and Supplementary Fig. 1). By contrast, *H. viridissima* had robust foot regeneration rates, with faster kinetics than *H. vulgaris* (Fig. 1j). *H. hymanae*, a member of another distinct group within the *Hydra* genus phylogeny, the Braueri clade, exhibited slightly reduced rates of foot regeneration, which varied depending on the bisection location. While 75% bisections resulted in nearly 100% successful foot regeneration, 25% bisections led to only ~60% regeneration (Fig. 1k). Overall, our survey of foot regeneration rates across the *Hydra* genus revealed variability that enables comparative studies. Furthermore, given the relatively high rate of foot regeneration in all species, except those in the Oligactis clade, we conclude that the common ancestor of the *Hydra* genus likely had robust rates of foot regeneration, which were reduced in the ancestor of the Oligactis clade.

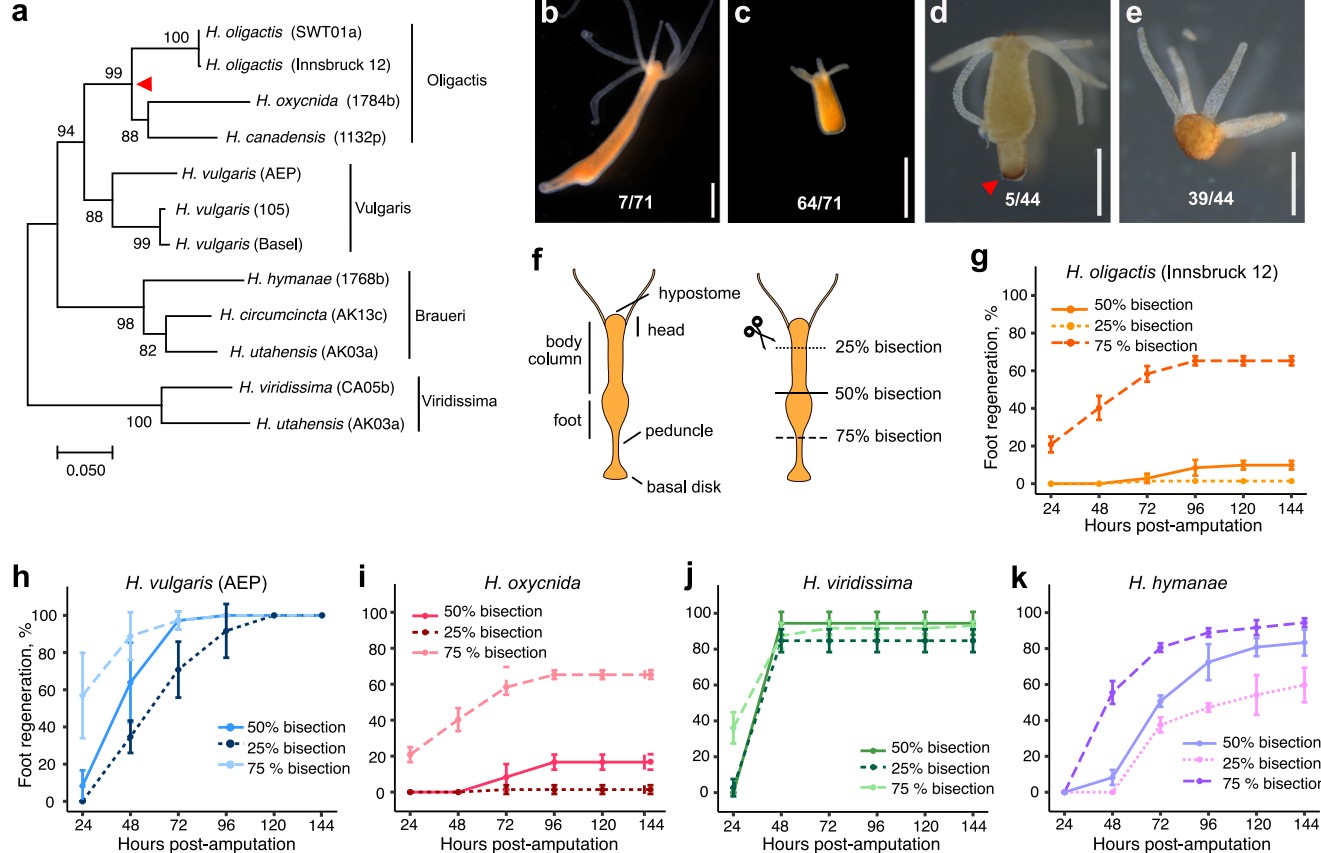

**Fig. 1 | Robust foot regeneration rates were lost in the Oligactis clade.**
**a** Maximum Likelihood phylogenetic tree of the *Hydra* genus based on mito-
chondrial cytochrome oxidase 1 sequences for well-characterized strains, selected
to represent the topology of the phylogenetic relationships within *Hydra*. The
genus is divided into four clades: Oligactis, Vulgaris, Braueri, and Viridissima[30]. A
red triangle marks the likely point of foot regeneration loss in the Oligactis clade.
All sequences and accession numbers used for this tree are provided in Source data
file 1. **b**, **c** Following 50% bisection, 90% of *H. oligactis* fail to regenerate their feet,
resulting in a stably footless phenotype. **d**, **e** Peroxidase staining correlates with
foot morphology in *H. oligactis*. The assay was performed on 44 *H. oligactis* polyps
9 days after 50% bisection (5 with feet, 39 footless), confirming the presence or
absence of a foot in all cases. Red triangle indicates peroxidase staining in the foot.

Scale bars: 1 mm. **f** Amputation strategy to evaluate foot regeneration abilities
across the *Hydra* genus. The top half of each bisected animal was visually mon-
itored for foot regeneration. Foot regeneration kinetics following amputations at
various positions along the oral-aboral axis for **g** *H. oligactis* (Innsbruck 12)
($n_{50\%} = 71$, $n_{25\%} = 72$, $n_{75\%} = 72$), **h** *H. vulgaris* (AEP) ($n_{50\%} = 72$, $n_{25\%} = 72$, $n_{75\%} = 72$),
**i** *H. oxycnida* (AUT08a) ($n_{50\%} = 72$, $n_{25\%} = 72$, $n_{75\%} = 72$), **j** *H. viridissima* (665a)
($n_{50\%} = 72$, $n_{25\%} = 72$, $n_{75\%} = 72$), and **k** *H. hymanae* (CA25a) ($n_{50\%} = 72$, $n_{25\%} = 72$,
$n_{75\%} = 72$). Three biological replicates of at least 22 animals were amputated for
each species and bisection level (total $n$ for each condition is listed above). Error
bars represent the standard deviation calculated from the foot regeneration per-
centage observed in each biological replicate. Foot regeneration data for all spe-
cies shown in this figure are provided in Source data file 2.

## Foot transcription fails to be activated in *H. oligactis*

To identify key aspects of the transcriptional response that result in
low foot regeneration rates in *H. oligactis*, we generated RNA-seq
libraries from the regenerating tips of bisected *H. oligactis* polyps
during a time course of both head regeneration and failed foot
regeneration at 0, 3, 12, 24, and 48 h post amputation (hpa) (Fig. 2a).
For the foot regeneration time course, it was not possible to determine
which samples would successfully regenerate their feet a priori,
therefore these samples contained a mixture of both successful and
unsuccessful regeneration. However, given the low rate of successful
foot regeneration after 50% bisection (~10%, Fig. 1g), most of the signal
is predicted to come from animals that will ultimately fail to regenerate
their feet. To define the transcriptional end point of successful
regeneration, we also collected RNA-seq data for head and foot tissue
in uninjured animals. We used these data to identify head- and foot-
specific genes in *H. oligactis* by determining the differentially expres-
sed genes in uninjured animals between the foot, head, and body
column tissue. The regenerating head and foot tissue at 0 hpa were
used as the body column gene expression signature for this analysis,
given that these tissues were taken just above and below the plane of
a 50% bisection right after injury (Fig. 2b). This approach uncovered

404 foot-specific genes and 987 head-specific genes in *H. oligactis*
(Fig. 2c, d and Supplementary Data file 1).

To analyze the transcriptional changes that occur over the course
of *H. oligactis* head regeneration and failed foot regeneration, we used
Principal Component Analysis (PCA) (Fig. 2e, f). In these analyses, we
included homeostatic head and foot RNA-seq libraries, which repre-
sent a successful regeneration end point. Notably, we found that *H.
oligactis* exhibits nearly 100% head regeneration after bisection, but
the process takes up to 72 hpa to complete. This is approximately 24 h
longer than in *H. vulgaris*, as determined morphologically by the pre-
sence of tentacles (Supplementary Fig. 2). Our RNA-seq time course of
*H. oligactis* head regeneration ended at 48 h, and our PCA analysis
confirms that head regeneration is not transcriptionally complete at
this point in *H. oligactis*. Although differences between homeostatic
head tissue and regenerating head tissue explain 83.34% of the varia-
tion among samples (PC1), regenerating head tissue exhibited a pro-
gressive upregulation of key PC1-associated genes, such as *wnt3*
(Supplementary Data File. 2), suggesting a shift toward head iden-
tity (Fig. 2e).

For failed foot regeneration, we did not observe a progressive
shift of regenerating tissue toward the homeostatic foot

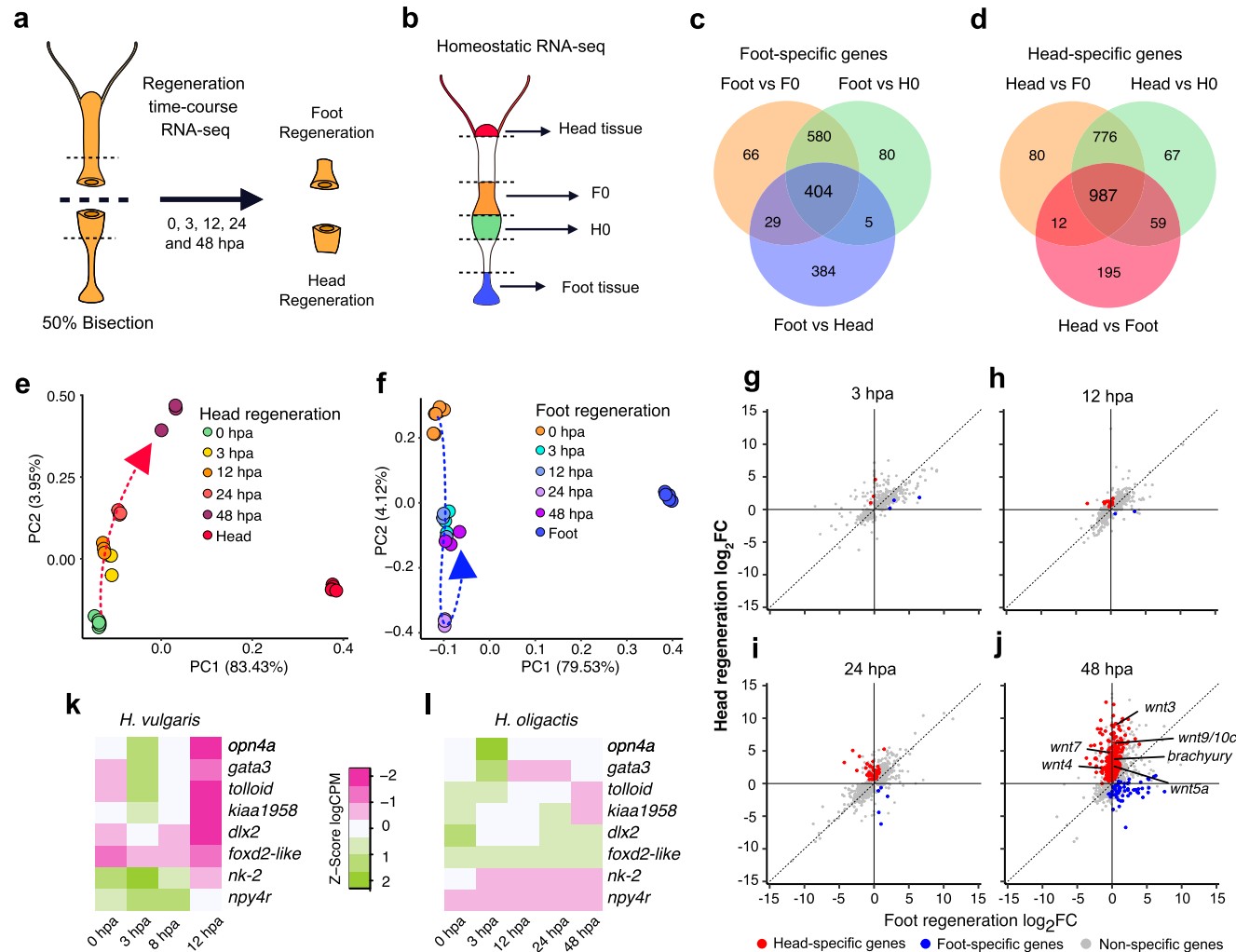

**Fig. 2 | Foot-specific gene transcription is largely absent during *H. oligactis* regeneration. a** RNA-seq library preparation strategy over a regeneration time course: Regenerating animals were incubated in 0.05% DMSO, these samples served as controls for the data shown in Fig. 5b. **b** RNA-seq strategy for different regions of homeostatic *H. oligactis*. F0 and H0 correspond to the 0 hpa foot and head regeneration samples in (**a**). **c, d** Venn diagrams showing the differential gene expression strategy used to identify 404 foot-specific genes (**c**) and 987 head-specific genes (**d**) in *H. oligactis*. Tissue-specific differentially expressed gene tables are provided in Source data file 3. **e** PCA plot of *H. oligactis* oral-facing wound gene expression during regeneration. **f** PCA plot of *H. oligactis* aboral-facing wound gene expression during regeneration. Log2 CPM used for PCA are provided in Source

data file 4. Comparison of log$_2$ fold change (log$_2$FC) transcript abundance between *H. oligactis* head regeneration and foot regeneration at (**g**) 3 hpa, (**h**) 12 hpa, (**i**) 24 hpa, and (**j**) 48 hpa. Red dots indicate differentially expressed head-specific genes at the oral-facing wound. Blue dots indicate differentially expressed foot-specific genes at the aboral-facing wound. Log2 fold change for time-course comparisons are provided in Source data file 5. **k** Heatmap of foot-specific gene expression in *H. vulgaris* (data from Cazet et al.)[19], showing upregulation by 12 hpa during foot regeneration. Values represent Z-scores calculated from log$_2$ counts per million (log$_2$CPM). **l** Heatmap of *H. oligactis* expression for the same 8 genes shown in (**k**). These genes do not exhibit rapid upregulation during regeneration. Log$_2$ CPM for selected foot-specific genes provided in Source data file 6.

transcriptional signature. Like head regeneration, the most variation (79.53%) for failed foot regeneration was explained by PC1, which denotes the difference in the expression profiles between homeostatic foot tissue and the regenerating samples. Importantly, 93 out of 100 of the topmost genes explaining variation within PC1 correspond to foot-specific genes (Supplementary Data file 3). Whereas only 4.12% of the variation is explained by PC2, which shows changes in the wounded tissue occurring during the time course of regeneration, but the regeneration time points did not change along the PC1 axis (Fig. 2f). This suggests that the gene expression profile of failed foot regeneration samples does not resemble that of the homeostatic foot at any timepoint.

We next examined the expression dynamics of head-specific and foot-specific transcripts over the time course of *H. oligactis* head regeneration and failed foot regeneration, respectively. To identify differentially expressed head and foot-specific genes, we compared

gene expression on each side of the wound at each time point. We found that in *H. oligactis* head regeneration, only a small number of head-specific genes showed tissue-specific upregulation at 12 hpa, with larger transcriptional changes becoming evident only after 24 hpa (Fig. 2g–j and Supplementary Data file 4 containing lists of tissue-specific genes at each timepoint). This contrasts with *H. vulgaris*, which shows significant structure-specific gene expression by 8 hpa for both head and foot regeneration[19]. For example, in *H. vulgaris*, several canonical Wnt pathway genes, including Wnt ligands, become specific to the oral wound by 8 hpa[19]. In *H. oligactis*, the expression of head-associated Wnt genes became specific to the oral wound by 48 hpa, and overall, 472 out of the 987 (47.8%) head-specific genes were expressed at this timepoint (Fig. 2j). In contrast, we identified only 63 out of 404 (15.6%) foot-specific genes as up-regulated in failed foot regenerating tissue at 48 hpa (Fig. 2g–j). Therefore, this analysis showed that by 48 hpa, head regeneration in *H. oligactis* although

incomplete, is progressing appropriately, whereas foot regeneration is largely failing at the transcriptional level.

To gain a deeper insight into the differences in foot regeneration outcomes between *H. vulgaris* and *H. oligactis*, we compared the injury-induced transcriptional activation of seven foot-specific genes that were identified in our previous transcriptional profiling of *H. vulgaris*. These genes were upregulated specifically at the aboral wound site by 12 hpa and were enriched in the foot as determined using the single cell expression atlas[19,31]; the transcript IDs for genes discussed in this study can be found in Supplementary Table 1). This analysis showed that transcriptional activation of these seven genes is weak in *H. oligactis* as compared to *H. vulgaris* (Fig. 2k–l). In addition, we compared the expression of foot-specific genes from *H. oligactis* to the expression of their orthologs in *H. vulgaris* (154 genes) and found lower expression in *H. oligactis* at every time point, including 0 hpa (Supplementary Fig. 3). The same comparison performed with a random gene set did not show expression differences (Supplementary Fig. 3). These data suggested that the initial foot competency of the *H. oligactis* tissue is lower (Supplementary Fig. 3).

### Injury-induced Wnt activation is reduced in *H. oligactis*

We next compared the early injury-induced transcriptional response between *H. vulgaris* and *H. oligactis* to identify differences that could lead to downstream failure of foot regeneration in *H. oligactis*. To accomplish this, we compared the *H. oligactis* regeneration RNA-seq data collected in this study to our previously published *H. vulgaris* regeneration RNA-seq data set, collected for both head and foot regeneration at 0, 3, 8, and 12 hpa[19]. To establish a framework to compare the gene expression patterns of *H. oligactis* and *H. vulgaris*, we used OrthoClust, a computational method that groups gene expression patterns based in gene orthology across multiple species into discrete clusters[32]. We identified orthologous genes by conducting a reciprocal BLAST analysis of the RNA-seq datasets for each species and from orthology prediction assignments obtained from a previous OrthoFinder analysis containing the predicted protein sequences of *H. oligactis, H. vulgaris*, and 42 addtional metazoan proteomes[23,33]. We identified 9371 transcripts for *H. oligactis* with a direct ortholog in *H. vulgaris*. We next used OrthoClust to compare the expression data of these transcripts over the course of regeneration to identify conserved co-expression modules across the two *Hydra* species. With this method, we identified nine gene co-expression clusters containing 566 transcripts from *H. vulgaris* and 489 transcripts from *H. oligactis* (Supplementary Data file 5).

While most of the OrthoClust modules contained transcripts that decreased in expression during the regeneration time course in both species, transcripts in clusters 5 and 6 showed an upregulation in head regenerating tissue in both *Hydra* species (Supplementary Fig. 4 and Fig. 3a–c). Cluster 6 contains several genes involved in canonical Wnt signaling. Notably, in *H. oligactis* head regeneration, cluster 6 genes such as *wnt3, wnt7*, and *brachyury* were delayed in their transcriptional activation, which correlates with the extended time frame of head regeneration in this species (Fig. 3d–i and Supplementary Fig. 2).

We found that some cluster 6 genes also behaved differently during foot regeneration when comparing the two species. In our previous study, we found that Wnt pathway genes are transiently upregulated from 0 to 3 hpa during *H. vulgaris* foot regeneration. This included the Wnt ligands *wnt3, wnt7*, and *wnt9/10 C* as well as the conserved Wnt signaling target *brachyury*[18–20] (Fig. 3d, f, h, j). By contrast, *wnt3, wnt7*, and *brachyury* were not significantly upregulated at 3pha in *H. oligactis* foot regeneration (Fig. 3e, g, i). Noteworthy, *wnt9/10c* is significantly upregulated in both *Hydra* species at 3 hpa during foot regeneration, although absolute levels of transcript are lower in *H. oligactis* (Fig. 3j, k). Altogether, these results revealed that injury-induced transcriptional activation of Wnt signaling genes is either absent or highly reduced in *H. oligactis* as compared to *H. vulgaris*.

### Reduced Wnt delays head organizer formation in *H. oligactis*

The hypostome in *Hydra* species has a conserved function as an oral organizer[34]. While it is almost certain that Wnt signaling plays a conserved role in directing axial patterning across the *Hydra* genus, the expression of Wnt ligands specifically in the hypostome has not been formally tested in *H. oligactis*. To address this, we performed fluorescent RNA in situ hybridization (FISH) to analyze the expression of *wnt3* in *H. oligactis*. We found that, like *H. vulgaris*, *wnt3* was expressed at the hypostome of *H. oligactis*, similar to *H. vulgaris* (Fig. 3l). Next, we used FISH to investigate the temporal and spatial expression of *wnt3* over the course of head regeneration. In contrast to *H. vulgaris*, the expression of *wnt3* was not detected by FISH in *H. oligactis* at 24 hpa. However, by 60 hpa, the expression of *wnt3* became apparent in *H. oligactis* (Fig. 3l, m). These results are consistent with slower head regeneration kinetics in *H. oligactis* (Supplementary Fig. 2) and suggest that delayed activation of Wnt signaling may contribute to differences in head regeneration timing between these species.

Given the morphological delay of head regeneration we observed in *H. oligactis* (Supplementary Fig. 2), along with the reduced levels of Wnt gene transcription (Fig. 3e, g, i, k, l), we hypothesized that formation of the oral organizer is delayed in *H. oligactis* as compared to *H vulgaris*. Tissue grafting is a classical method for determining the timing of head organizer formation: regenerating head tissue is grafted onto the body column of a host animal to test if the donor tissue can induce a secondary axis. Using this approach, previous studies demonstrated that regenerating *H. vulgaris* head tissue acquires organizing ability as early as 8 hpa, which also correlates with high levels of Wnt signaling pathway gene transcripts at this timepoint[19,35]. To test the timing of head organizer formation in *H. oligactis*, we conducted classic grafting experiments. We also performed these experiments in *H. vulgaris* as a positive control. We bisected *Hydra* polyps and grafted the tissue from the oral injury site onto a host polyp at 0, 3, 8, 16, 24, and 48 hpa (Fig. 3n). We then quantified the frequency of secondary axis formation at 5 days post grafting (Fig. 3o). While we observed organizer ability in regenerating *H. vulgaris* head tissue as early as 8 hpa, as expected, regenerating *H. oligactis* did not begin showing organizer activity until 16 hpa. *H. oligactis* head organizing ability was reduced as compared to *H. vulgaris* until 48 hpa (two-way ANOVA, *p* value = $2.55 \times 10^{-5}$, F-value = 23.91, df = 1), at which time a similar capacity to direct secondary axis formation in both species was observed.

Our RNA-seq, FISH, and grafting experiments support the conclusion that injury does not strongly activate Wnt signaling during the early stages of regeneration in *H. oligactis*, which contributes to slower head regeneration kinetics. To further test this, we asked whether pharmacological activation of Wnt signaling could accelerate head regeneration. As a first step, we examined whether the Wnt agonist Alsterpaullone (ALP), which reliably induces Wnt signaling through β-catenin stabilization in *H. vulgaris* and produces ectopic tentacles along the body column[17,36], elicited a similar phenotype in *H. oligactis*. Uninjured *H. oligactis* were treated with 5 μM ALP for 24 h, then washed with *Hydra* medium. Ectopic tentacle formation was evident as early as 24 h after washout (Supplementary Fig. 5). This experiment validates the use of ALP in *H. oligactis* to activate Wnt signaling.

We next tested whether ALP-induced Wnt activation could accelerate head regeneration in *H. oligactis*. Treating the lower halves of 50% bisected animals with 5 μM ALP for 3 hpa enhanced regeneration kinetics, with the proportion of animals showing tentacle formation by 48 h increasing from about 20% to over 60%, a level more comparable to *H. vulgaris* (Supplementary Figs. 2 and 6). Notably, though, approximately 30% of treated animals also developed ectopic tentacles at various positions along the body column, consistent with

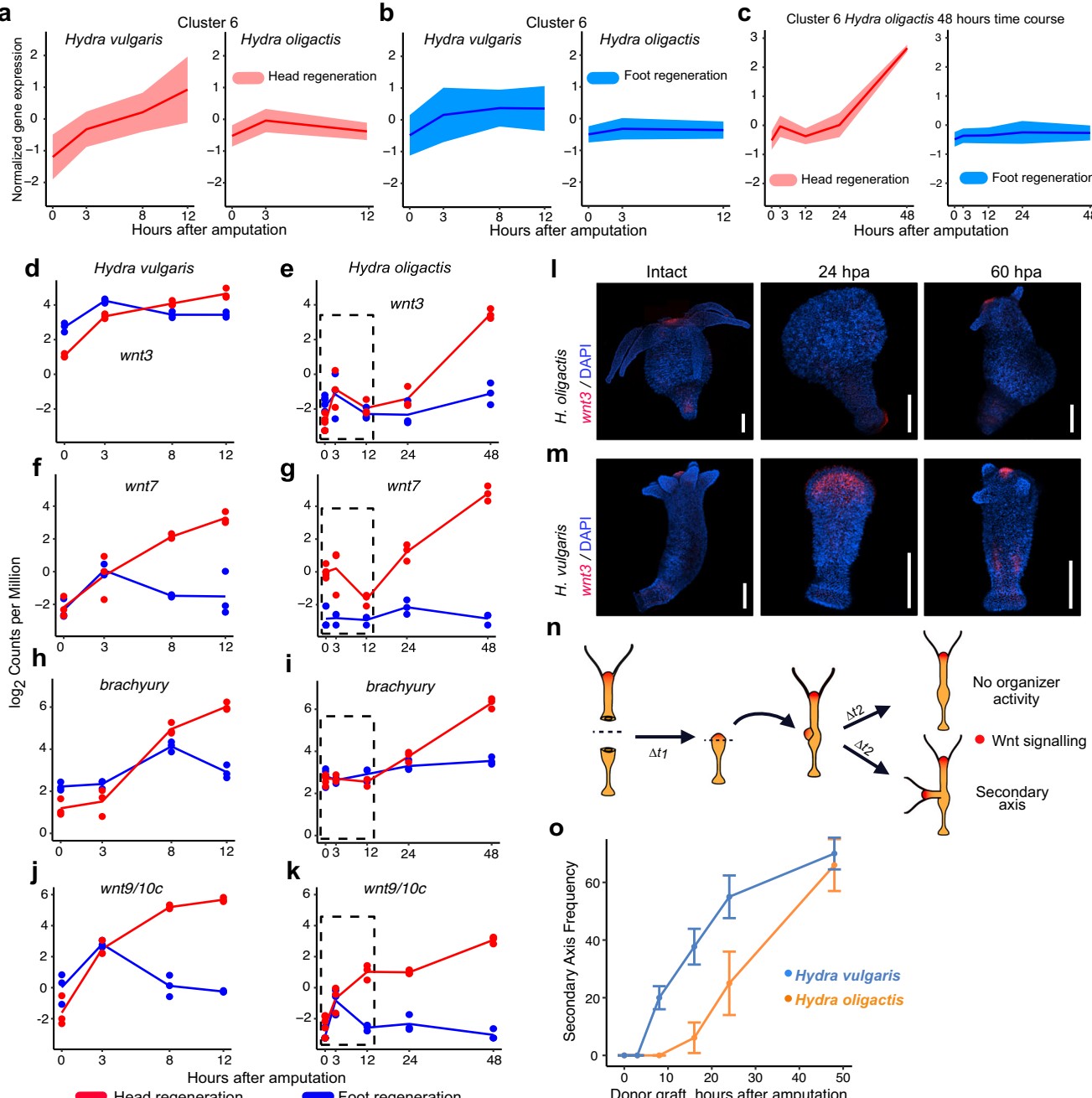

**Fig. 3 | Delayed transcriptional activation of Wnt signaling genes during regeneration in *H. oligactis* as compared to *H. vulgaris*.** Normalized expression patterns for genes in cluster 6, identified with OrthoClust, in (**a**) *H. vulgaris* and (**b**) *H. oligactis* over the first 12 h of regeneration. Expression patterns at the oral-facing wound are shown in red, while aboral-facing wounds are shown in blue. **c** Normalized expression patterns for cluster 6 in the extended regeneration time course in *H. oligactis*. The lighter colored shadow in ribbon plots represents the standard deviation of FPKM at each time point. FPKM for OrthoClust analysis is provided in Source data file 7 for *H. oligactis* and Source data file 8 for *H. vulgaris*. **d–k** RNA-seq expression profiles for (**d, e**) *wnt3*, (**f, g**) *wnt7*, (**h, i**) *brachyury* (*bra1*) and (**j–k**) *wnt9/10 C*, during head (red) and foot (blue) regeneration in *H. vulgaris* (data from Cazet et al.)[19] and *H. oligactis* (this study, see Fig. 2). Note the extended time course for *H. oligactis*, the dashed box encloses the expression time window that can be directly compared to *H. vulgaris* data. Normalized log2 CPM are provided in Source data file 4 for *H. oligactis* and Source data file 9 for *H. vulgaris*. **l-m** Confocal FISH images for *wnt3* in homeostatic and regenerating tissue in (**l**) *H. oligactis* and (**m**) *H. vulgaris* at 0, 24, and 60 hpa. Scale bars: 500 μm. RNA in situ hybridization experiments were performed twice, with 20 animals displaying similar patterns per probe and timepoint. Blue shows DAPI, and red shows the RNA probes. **n** Experimental strategy to evaluate head organizer activity during head regeneration. **o** Secondary axis induction following grafting of injured oral tissue onto host animals at 0, 3, 8, 16, 24, and 48 hpa. The blue line represents *H. vulgaris* and the orange line *H. oligactis*. Two-way ANOVA was performed on this time series between species (*p* value = $2.55 \times 10^{-5}$, F-value = 23.91, df = 1). Error bars represent the standard deviation from three biological replicates (*n* = 30). Given the large number of timepoints the details of sample sizes and secondary-axis formation are provided in Source data File 10.

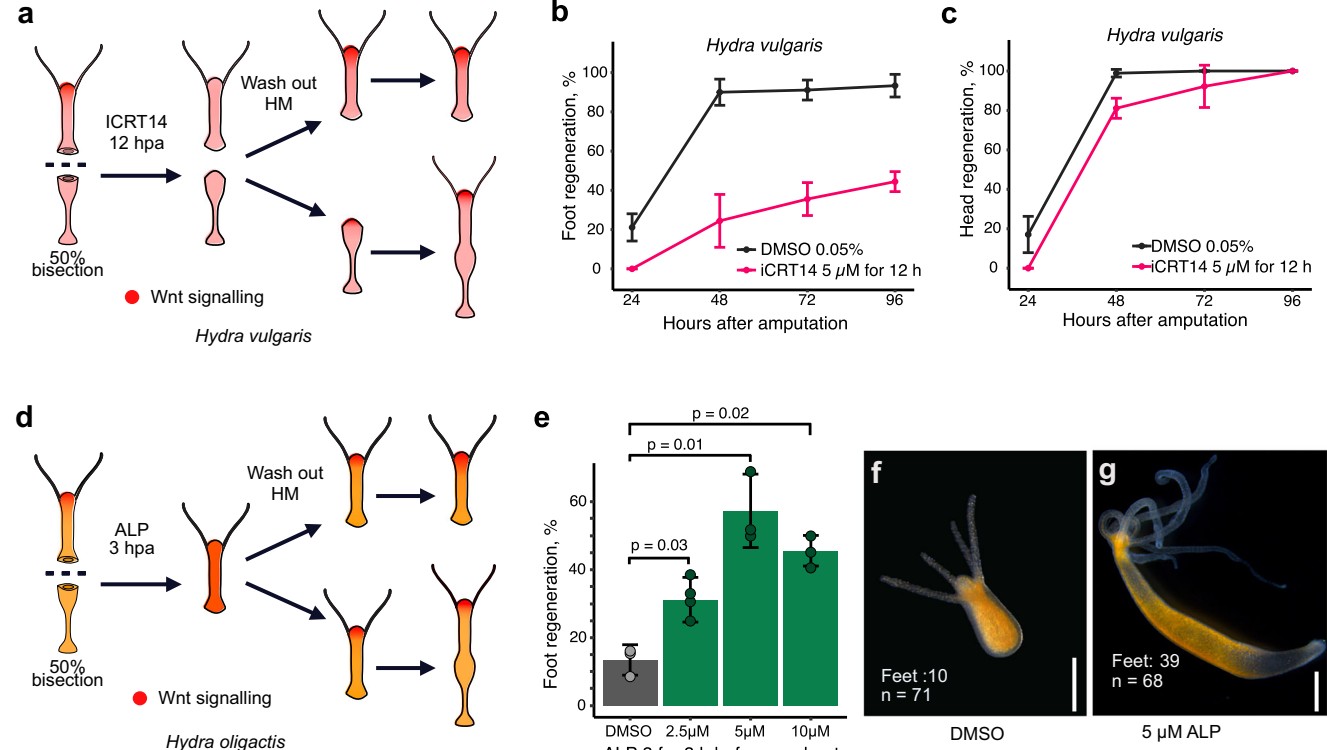

**Fig. 4 | Transient activation of Wnt signaling increases rate of foot regeneration in *H. oligactis*. a** Experimental strategy to assess foot and head regeneration in *H. vulgaris* after treatment with the Wnt inhibitor iCRT14. **b** Line graph showing foot regeneration kinetics of three different biological replicates composed of at least 29 *H. vulgaris* each, treated with iCRT14 followed for 96 hpa (pink; batch sizes, $n = 30$, $n_2 = 30$, $n_3 = 30$) and DMSO controls (black; batch sizes, $n_1 = 29$, $n_2 = 30$, $n_3 = 30$). Error bars represent the standard deviation calculated from each biological replicate. **c** Line graph showing head regeneration kinetics in *H. vulgaris* treated with iCRT14 (pink; batch sizes: $n_1 = 29$, $n_2 = 30$, $n_3 = 30$) and DMSO controls (black; batch sizes, $n_1 = 29$, $n_2 = 29$, $n_3 = 29$). Error bars represent the standard deviation calculated from the different biological replicates. Foot and head regeneration percentages for iCRT14 treated *H. vulgaris* are provided in Source data file 11. **d** Experimental strategy for treating 50% bisected *H. oligactis* with the Wnt agonist Alsterpaullone (ALP). **e** Bar graph showing the percentage of successful

foot regeneration in *H. oligactis* following a 3-h post-bisection treatment with DMSO (batch sizes: $n_1 = 26$, $n_2 = 24$, $n_3 = 24$) or ALP at 2.5 μM (batch sizes: $n_1 = 21$, $n_2 = 21$, $n_3 = 23$, $n_4 = 24$), 5 μM (batch sizes: $n_1 = 23$, $n_2 = 23$, $n_3 = 23$) and 10 μM (batch sizes: $n_1 = 24$, $n_2 = 22$, $n_3 = 24$). Green bars show ALP concentrations treatments, grey bar is DMSO control. Statistical comparisons of DMSO versus ALP treatments were conducted using two-tailed t-tests. For 2.5 μM ALP, the *p* value was 0.0214 ($t = -6.73$, df = 2; 95% CI: −29.05 to −6.38; mean difference = −17.72). For 5 μM ALP, the p-value was 0.0154 ($t = -7.97$, df = 2; 95% CI: −67.42 to −20.15; mean difference = −43.78). For 10 μM ALP, the *p* value was 0.0214 ($t = -6.73$, df = 2; 95% CI: −48.15 to −10.58; mean difference = −29.37). Error bars represent the standard deviation calculated from each biological replicate per condition. **f** Representative image of *H. oligactis* showing failed foot regeneration in DMSO-treated control group. **g** Representative image of *H. oligactis* showing successful foot regeneration after treatment with 5 μM ALP. Scale bars: 500 μm.

the broad effects of ALP treatment. Together, these results indicate that ALP robustly induces Wnt signaling in *H. oligactis* and further support the conclusion that the slower head regeneration kinetics in this species arise, at least in part, from low injury-induced Wnt activation.

## Wnt inhibition in *H. vulgaris* mimics *H. oligactis* phenotype

A previous study showed that treatment with iCRT14 inhibits both head and foot regeneration in *H. vulgaris* upon continuous treatment over the course of regeneration[20]. Given our findings that in *H. oligactis* the injury response does not elicit strong expression of Wnt signaling genes, here we sought to mimic that effect in *H. vulgaris* by transiently treating animals with iCRT14 for 12 h after bisection (Fig. 4a). This treatment caused *H. vulgaris* regeneration to more closely resemble *H. oligactis* regeneration. Specifically, we found that transient inhibition of Wnt signaling in *H. vulgaris* resulted in reduced foot regeneration rates (two-way ANOVA, *p* value = 0.002824, F-value = 4.99, df = 12) and a slight delay in head regeneration (Fig. 4b, c and Supplementary Fig. 7). Notably, this treatment produced stably footless *H. vulgaris* mimicking the *H. oligactis* phenotype (Supplementary Fig. 7). Our results support the conclusion that a lack of injury-induced Wnt signaling significantly contributes to foot regeneration deficiency in *H. oligactis*.

## Wnt activation improves foot regeneration in *H. oligactis*

Considering our findings and previous studies showing that treatment with a Wnt inhibitor negatively impacts *H. vulgaris* foot regeneration[20]. We next asked how injury-induced Wnt expression impacts foot regeneration abilities in *Hydra*. We hypothesized that the lack of injury-induced Wnt expression contributes to low rates of *H. oligactis* foot regeneration. To test this, we pharmacologically activated Wnt signaling during the injury phase of *H. oligactis* foot regeneration using ALP. We transiently treated the upper halves of 50% bisected *H. oligactis* with different concentrations of ALP (2.5, 5, and 10 μM) for 3 h after injury, to mimic the short window of Wnt signaling activation in *H. vulgaris* foot regeneration. We then monitored the animals over 6 days for the presence or absence of foot regeneration (Fig. 4d). Treatment of regenerating *H. oligactis* with ALP, regardless of the concentration, resulted in a significant increase of foot regeneration rates, with the concentration of 5 μM having the largest effect (Fig. 4e–g). Notably, at 5 μM, ~7% of animals regenerated a second head at the aboral wound site instead of a foot, with this percentage increasing to 20% at 10 μM ALP (Supplementary Fig. 8). The appearance of the two-headed phenotype suggests that a precise balance in Wnt activation is required to determine whether foot or head regeneration occurs at the aboral wound site. In addition, the higher occurrence of two-headed *Hydra* in

the 10 μM ALP treatment likely explains the reduced number of animals that regenerated a foot under this concentration.

Given that our results suggest that transient Wnt activation is required for foot regeneration in *H. oligactis*, we tested if inhibition of Wnt/β-catenin signaling by iCRT14 treatment would further reduce its foot regenerative ability. We found that iCRT14 treatment for the first 12 h after injury results in a slightly lower rate of foot regeneration in *H. oligactis* (Supplementary Fig. 9). Overall, these results support the conclusion that a transient burst of Wnt signaling activation during the generic injury response is essential for foot regeneration in *Hydra*.

### Wnt activation induces foot gene expression in *H. oligactis*

Our finding that transient activation of Wnt signaling increases foot regeneration rates in *H. oligactis* is consistent with published literature showing that injury-induced Wnt activation plays a role in *Hydra vulgaris* foot regeneration[19,20,37]. However, the mechanisms by which Wnt signaling functions in foot regeneration is unknown. To shed light on this, we generated RNA-seq libraries from a time course of *H. oligactis* foot regenerating fragments after a 3-h treatment with 5 μM ALP (Fig. 5a). These data revealed that ALP treatment elicited the transient upregulation of Wnt pathway genes such as *tcf*, *wnt3*, *axin1*, *wntless* and *sFRP3*, demonstrating that ALP treatment transcriptionally activated Wnt signaling as expected (Supplementary Fig. 10). Furthermore, PCA of ALP-treated (rescued) and DMSO-treated (control) samples revealed that by 48 hpa, transient activation of Wnt promoted transcriptional changes in the direction of the homeostatic foot along the PC1 axis (Fig. 5b).

To characterize the transcriptional changes elicited by ALP treatment, we performed differential gene expression analysis comparing ALP-treated and control samples. We found that Wnt pathway activation induced the transcriptional activation of foot-specific genes, with 52 and 134 foot-specific genes being upregulated in ALP treatment conditions as compared to control conditions by 24 hpa and 48 hpa, respectively (Fig. 5c, d). In contrast, just a few foot-specific genes were upregulated in control conditions as compared to ALP treatment conditions at these same timepoints (Fig. 5c, d).

To improve our understanding of the mechanisms by which transient ALP treatment promotes foot regeneration, we used maSig-Pro, an R package that analyses time course expression data in different conditions through a regression approach to identify differential gene expression patterns[38]. This analysis uncovered 1773 genes with differential expression patterns grouped into 9 modules (Fig. 5e and Supplementary Data file. 6). The expression pattern of module 5 showed upregulation at 3 hpa only in ALP-treated tissue, followed by downregulation at later timepoints, and included Wnt signaling pathway genes (*tcf*, *axin1*, *lgr5*, *sp5*, and *nphp3*), confirming the validity of this method. In addition, several transcription factors (TFs) whose orthologs are involved in cell differentiation and development were included in module 5, such as *foxl1*, *dmrta2*, *ywhaz*, *aatf*, *rfx1*, and *tle3*. These TFs are also expressed in the homeostatic *Hydra* but are not foot-specific. To better understand the functions related to each module, we performed Gene Ontology (GO) term enrichment analysis for all modules (Supplementary Data file 7). Module 5 was enriched for gene expression regulatory functions, including transcription factors and coactivators. Contrastingly, module 2 showed downregulation in response to ALP at 3 hpa of transcriptional regulators involved in the generic injury response, such as bZIP TFs *creb*, *jun*, and *fos*[19,39] (Fig. 5e and Supplementary Fig. 11).

We also identified modules with foot-specific genes upregulated in the ALP-treated samples at 12 hpa (module 4) and 24 hpa (module 7) (Fig. 5e–g). GO analysis of module 7 showed enrichment for morphogenesis-related functions (Fig. 5g), including β-catenin and genes in the BMP and Notch pathways. Importantly, among the foot-specific genes we found several TFs that were significantly up-regulated in response to Wnt activation by ALP treatment,

particularly *dlx2*, which was also included in module 7, was upregulated at 12 hpa in response to ALP, making it the earliest foot-specific TF to be transcriptionally activated upon Wnt activation (Fig. 5h–i). Noteworthy, a previous study showed that *dlx2* is essential for foot regeneration in *H. vulgaris*, highlighting its likely role as a key regulator of foot regeneration[40]. Since *dlx2* was the earliest foot-specific transcription factor activated by ALP, we used FISH to examine its spatial expression in *H. oligactis* and *H. vulgaris*. In uninjured *H. oligactis*, *dlx2* expression was restricted to a small peduncle region, whereas in *H. vulgaris*, it extended to the basal disk (Fig. 5j–m), suggesting differences in *dlx2* regulation under homeostatic conditions. At 24 hpa, *dlx2* was clearly expressed at the aboral injury site in *H. vulgaris* but was undetectable in *H. oligactis* (Fig. 5l–m), reinforcing its correlation with foot regeneration outcomes.

GO analysis of module 4 revealed enrichment for terms related to extracellular matrix organization (Fig. 5f). Notably, Wnt signaling has been found to induce extracellular matrix remodeling as part of the tissue patterning that occurs during *H. vulgaris* regeneration[41]. In addition, this module included TFs *gata3* and *nk2*, previously identified as foot-specific in *H. vulgaris*, as well as other TFs that are upregulated temporarily at 24 hpa with ALP treatment (Supplementary Fig. 12). Overall, these findings highlight the hierarchical gene regulatory network underlying foot regeneration in *Hydra*, where transient Wnt activation is required to trigger a transcriptional cascade that sequentially activates non-tissue specific TFs, followed by foot-specific TFs, starting with *dlx2*, and ultimately genes involved in morphological changes such as extracellular matrix remodeling, leading to successful tissue regeneration.

## Discussion

An important finding of our study is that transient, injury-induced Wnt signaling is essential for foot regeneration in *Hydra*, and that changes in the strength of this induction can alter foot regeneration rates. Previous studies in planarians have likewise shown that differences in Wnt signaling strength affect regenerative outcomes[4–7]. Together, these findings suggest that evolutionary gains and losses of regenerative ability could arise through modulation of Wnt signaling levels. While Wnt signaling is required for oral patterning in *Hydra*, it functions in posterior patterning in planarians[3]. In both planarians and *Hydra*, regeneration defects arise in tissue opposite the high Wnt pole: head regeneration defects are common in planarians, whereas foot regeneration is impaired in *H. oligactis*. In planarians, head regeneration defects are linked to excessive Wnt signaling activation and can be rescued by Wnt inhibition, for example, through downregulation of β-catenin[4–7]. By contrast, we found that low foot regeneration rates in *H. oligactis* result from insufficient injury-induced Wnt activation, and that foot regeneration rates can be increased by transiently boosting Wnt signaling with an agonist. Thus, although altered Wnt dynamics underlie regeneration defects in both *Hydra* and planarians, the mechanisms differ: in planarians, excess homeostatic Wnt disrupts head regeneration by interfering with axial patterning, while in *Hydra*, a distinct injury-induced Wnt signal, separate from its well-established role in head patterning, is required for foot regeneration. These findings point to a two-phase model of Wnt signaling in *Hydra*, in which an early injury-induced Wnt burst is necessary for initiating foot regeneration through mechanisms that remain to be clearly defined, followed by the later role of Wnt in axial patterning at the oral end. Our data also reveal the need of a balance at the aboral end, where excessive activation of the transient injury-induced Wnt signal can shift the system toward head formation, producing animals with heads at both poles, a phenotype also reported in *H. vulgaris*[37]. Future studies are needed to clarify how Wnt signaling functions in these distinct contexts, both in rendering tissue competent for foot regeneration and in tipping the balance toward head regeneration.

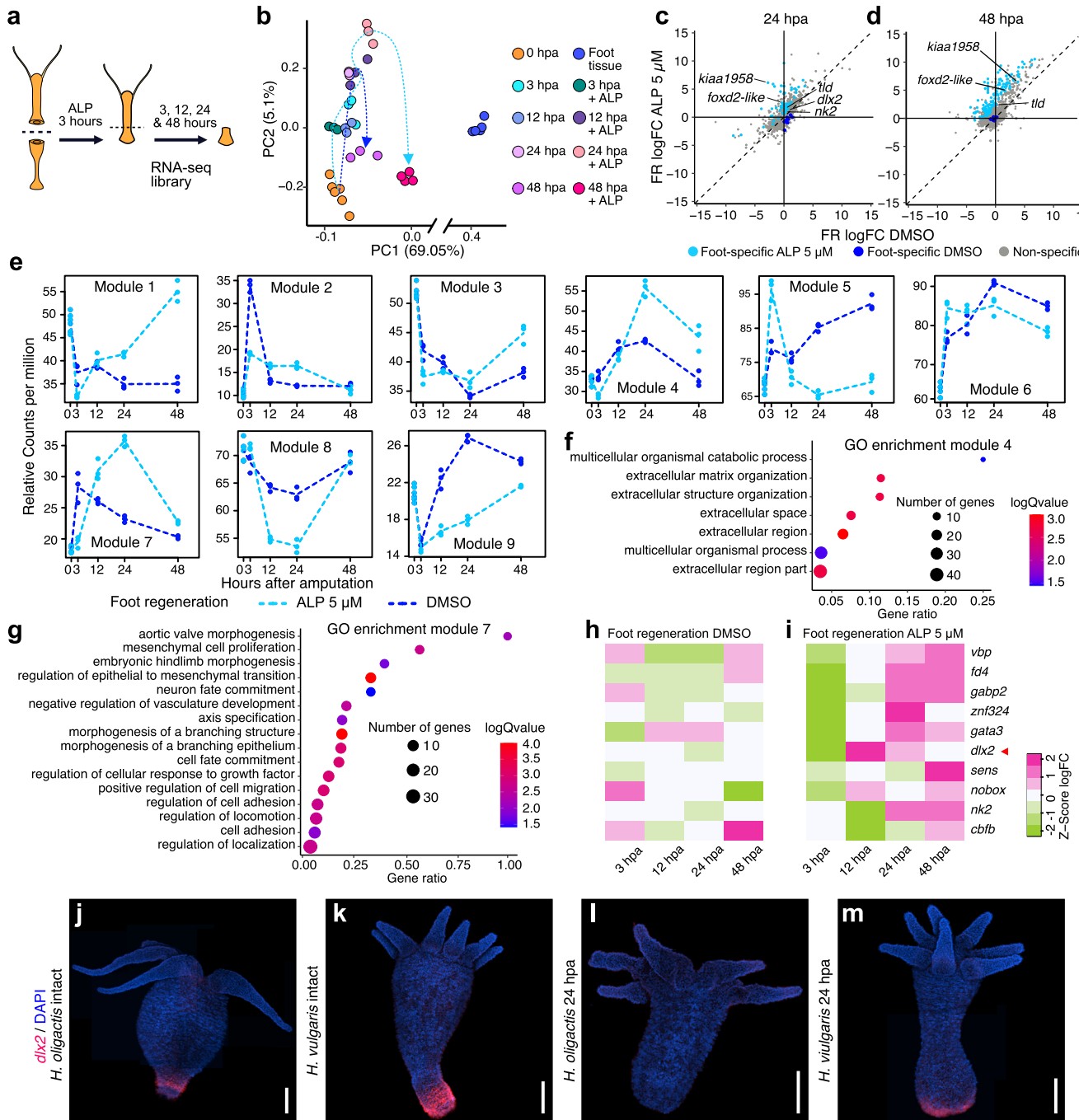

**Fig. 5 | Wnt-dependent activation of the foot regeneration transcription factor *dlx2*. a** Experimental strategy for producing RNA-seq libraries from ALP-treated aboral-facing wound tissue. **b** PCA plot showing transcriptional trajectories during regeneration in DMSO-treated (blue dotted line; data from Fig. 2) and ALP-treated samples (green dotted line) Note the broken *X* axis. ALP and DMSO-treated log₂ CPM data for foot regenerating *H. oligactis* are provided in Source data file 13. Comparison of average log₂ fold change (log₂FC) in transcript abundance between failed foot regeneration (DMSO-treated control animals) and rescued foot regeneration (ALP-treated animals) at (**c**) 24 hpa and (**d**) 48 hpa. Blue dots represent foot-specific genes enriched in failed foot regeneration tissue; light blue dots represent foot-specific genes enriched in ALP-treated rescued foot regenerating tissue. Fold change gene tables for foot regenerating *H. oligactis* in DMSO and ALP treatments are provided in Source data file 14. **e** Gene expression modules for failed foot

regeneration (DMSO-treated control animals, blue dashed line) and rescued foot regeneration (ALP-treated animals, light blue dashed line) as identified and plotted by maSigPro. **f** Gene Ontology enrichment analysis for genes in module 4. **g** Gene Ontology enrichment analysis for genes in module 7. GO enriched terms for modules 4 and 7 are provided in Source data file 15. Heat maps showing log fold change of foot-specific transcription factors identified here during foot regeneration under control (**h**) and ALP-treated (**i**) conditions. The red triangle highlights *dlx2* expression changes over time. Log fold change values for both heatmaps are provided in Source data file 16. **j**–**m** Confocal images of RNA in situ hybridizations for *dlx2* in untreated *H. oligactis* and *H. vulgaris*, showing intact polyps and aboral regenerating tissue at 24 hpa. RNA in situ hybridization experiments were performed twice, with 20 animals displaying similar patterns per condition per timepoint. Blue shows DAPI, and red shows the RNA probes. Scales bars: 500 μm.

The molecular basis for weaker injury-induced Wnt signaling in *H. oligactis* compared to *H. vulgaris* remains unclear. However, one possibility is the presence of a stronger Wnt inhibitory environment in *H. oligactis*. In *Hydra*, it is hypothesized that the head organizer establishes a morphogenetic gradient through a locally self-reinforcing Wnt signal and a secreted long-range inhibitory signal, which determines cell fate along the oral-aboral axis[9,42]. Previous studies suggest that this head inhibitory signal is a secreted Wnt antagonist[43,44], though its precise identity remains unknown. Notably, prior studies proposed that the reason for limited foot regeneration in *H. oligactis* is higher levels of free head inhibitor compared to *H. vulgaris*[15,45]. If this inhibitor is indeed a Wnt antagonist, this would be consistent with our findings that injury-induced Wnt signaling is attenuated in *H. oligactis*. In addition, we found that foot regeneration potential in *H. oligactis* increases with greater distance from the head, further supporting the hypothesis that a Wnt-inhibitory gradient emanating from the head influences the strength of injury-induced Wnt induction and, consequently, foot regenerative capacity.

In addition to a strong Wnt inhibitory environment in *H. oligactis*, reduced transcriptional activation of injury-induced Wnt signaling genes may also contribute to its impaired foot regeneration. In *H. vulgaris*, bZIP TFs are implicated in activating the expression of Wnt signaling genes during the general injury response[19,37]. Although *H. oligactis* demonstrates injury-induced transcription of bZIP TFs comparable to *H. vulgaris* (Supplementary Fig. 11), it is possible that in *H. oligactis*, bZIP TFs fail to activate the transcription of Wnt genes due to reduced protein stability or weaker binding affinity to Wnt gene regulatory regions. Supporting this, previous research revealed stronger CREB-binding activity in nuclear extracts from *H. oligactis* injured oral tissue compared to injured aboral tissue, whereas *H. vulgaris* showed equal CREB binding activity at both wound sides[46]. Moreover, our data indicate regulatory feedback between Wnt signaling and bZIP TFs. ALP treatment causes downregulation of bZIP TFs at 3hpa in *H. oligactis*, consistent with the extended bZIP transcriptional activation observed in *H vulgaris* when inhibiting Wnt signaling with iCRT14[19]. This suggests a bidirectional regulatory relationship between bZIP TFs and Wnt signaling. Future research is necessary to investigate if differences in bZIP TF-mediated transcriptional activation of Wnt signaling genes contribute to the *H. oligactis* foot regeneration defect and how this process may be influenced by the stronger Wnt inhibitory gradient proposed for this species.

In our previous study, we found that prolonged injury-induced Wnt signaling is sufficient to induce ectopic head formation[19]. However, our current findings reveal that injury-induced Wnt signaling activation is not strictly required for head regeneration. Transient inhibition of Wnt signaling activation during the injury phase of regeneration delays, but does not block *H. vulgaris* head regeneration, mirroring the extended timeline we observed for *H. oligactis* head regeneration. Our findings suggest two potential, non-mutually exclusive mechanisms for establishing the Wnt organizer during *H. oligactis* head regeneration. First, the weak injury-induced expression of *wnt9/10 C* in *H. oligactis* suggests that while injury-triggered Wnt activation may contribute to head organizer formation, the associated feedforward loop could take longer to establish due to the low initial signal. Second, an unidentified mechanism may activate *Wnt* gene transcription later during head regeneration, either as the sole mechanism in *H. oligactis* or in addition to weak injury-induced Wnt activation. A similar late-acting mechanism may also function in *H. vulgaris*, enabling head regeneration even when injury-induced Wnt activation is inhibited (Fig. 4c).

Consistent with a prior study showing that knockdown of the TF Dlx2 inhibits foot regeneration[40], placing *dlx2* as a key regulator of this process, our findings indicate that *dlx2* is the first TF expressed during rescued foot regeneration in *H. oligactis*, further supporting its function as a high-level regulator of the process. Importantly, we demonstrate that *dlx2* expression is dependent on treatment with the Wnt agonist. Thus, while our findings confirm the essential role of *dlx2* in foot regeneration across the *Hydra* genus, they also provide new evidence linking its activation to Wnt signaling. In this same vein, the TF Gata3 has also been identified as a positive regular of basal disk fate[47], and may be downstream of *dlx2*, based on its expression dynamics in *H. oligactis* ALP-rescued foot regeneration. However, *Hydra* possess two epithelial layers, the endoderm and ectoderm, which must interact to specify the foot[48]. Interestingly, *dlx2* and *gata3* are both expressed in the ectoderm[31], whereas *nk2*, another foot-specific TF, is expressed in the endoderm[49]. Although *nk2* has been proposed as a regulator of foot specification[49], functional evidence remains lacking. Our study shows that *dlx2* is upregulated by 12 hpa in ALP-treated foot regenerating tissue, whereas *nk2* upregulation occurs at 24 hpa, suggesting that ectodermal specification may occur before endodermal specification. Future studies should investigate how Wnt signaling activates *dlx2* in the ectoderm and how ectodermal and endodermal regulatory programs interact during foot regeneration. Constructing tissue-specific gene regulatory networks will be essential to understanding how Wnt signaling coordinates regeneration across both epithelial layers.

Our findings suggest that robust foot regeneration ability is ancestral in the *Hydra* genus and that reduced foot regeneration rates are a derived trait of the Oligactis group. Unlike most *Hydra* species, which produce gametes continuously throughout life, Oligactis species exhibit semelparity, a reproductive strategy in which individuals produce a large number of gametes before dying[50]. In *H. oligactis*, exposure to low temperatures induces gametogenesis, leading to somatic senescence through stem cell exhaustion[51,52]. One possibility is that reduced foot regeneration in the Oligactis clade evolved as a trade-off associated with semelparous reproduction. Such trade-offs between regenerative capacity and sexual reproduction have been previously proposed[10] in the semelparous planarian *Procotyla fluviatilis*, which displays a head regeneration defect that can be rescued by inhibiting Wnt signaling[6]. In planarians, it is proposed that increased Wnt signaling blocks head regeneration but exerts a beneficial effect on sexual reproduction[4]. By contrast, in *H. oligactis*, we observed that low injury-induced Wnt activation is a cause of low foot regeneration rates, suggesting that in this case, sexual reproduction benefited from low Wnt signaling. Similar trade-offs involving Wnt signaling may occur in other animals. For example, sexually mature male zebrafish show impaired regeneration of amputated pectoral fins due to androgen-driven inhibition of Wnt signaling, which can be reversed by pharmacological activation of Wnt signaling[53]. These examples suggest that adaptations modulating reproduction may compromise regenerative capacity, highlighting the need for further research on how Wnt signaling mediates the trade-off between reproductive strategies and regeneration in *Hydra* and other animals.

While Wnt signaling activation during regeneration is a shared feature among distantly related animals, the gene regulatory networks linking early injury response to Wnt signaling activation are not entirely conserved across taxa[1]. Such variation limits our ability to draw broad conclusions about the evolutionary gain or loss of regenerative abilities, underscoring the need for more comparative studies. However, comparisons across long evolutionary distances can be difficult to interpret due to the distinct biological contexts in which these molecular pathways function. For this reason, studies of closely related animals with varying regenerative abilities, such as previously published studies in planarians and the one presented here, provide valuable insights into specific evolutionary rewiring events in the gene regulatory networks underlying regeneration. Such research can reveal the molecular changes driving the diversity of regenerative capacity observed across animals. By identifying these evolutionary changes, we may uncover the molecular requirements necessary to induce regeneration in animals with limited regenerative potential.

## Methods

### Hydra strains and culture

Here, we used two strains of *H. oligactis*. The Cold Resistant Swiss strain provided by Brigitte Galliot was used only in regeneration experiments shown in Supplementary Fig. 1[51]. The second strain used for all remaining *H. oligactis* experiments was collected in Innsbruck, Austria, and adapted to laboratory conditions by Bert Hobmayer[19]. *H. vulgaris* AEP strain was used for comparative approaches throughout this work. The remaining *Hydra* species used for the regeneration experiments shown in Fig. 1: *H. oxycnida* (AUT08a, Austria), *H. hymanae* (CA25a, USA), and *H. viridissima* (665a, Peru), were collected and provided by Ruthie B. Spencer, Daniel E. Martínez, and Robert E. Steele. The taxonomical classification of these *Hydra* species was validated by sequencing a fragment of nuclear ribosomal DNA that included 18S (partial), the internal transcribed spacer 1 (ITS; complete), 5.8S (complete), ITS2 (complete) and 28S partial, amplified using primers designed to anneal to the 18S (5′-CACCGCCCGTCGCTACTACCGATTGAATGG-3′) and 28S (5′-CCGCTTCACTCGCCGTTACTAGGGGAATCC-3′) ribosomal genes. All *Hydra* species were kept in *Hydra* medium (0.38 mM CaCl₂, 0.32 mM MgSO₄ X 7H2O, 0.5 mM NaHCO₃, 0.08 mM K₂CO₃), and fed three days per week with Brine Shrimp during experimentation. All animals used in this study were kept in their asexual stage; no sex induction was performed, given that the conditions required to induce sexual differentiation in *Hydra* are impractical and not required for the observation of regeneration.

### Phylogenetic analysis

The taxonomy of the *Hydra* genus has already been reported in detail[30]. To show the topology of the tree for the species used in this study (Fig. 1a), we selected representative validated species from each major group in the *Hydra* genus phylogeny, and the sequence for CO1 was used to build a maximum likelihood tree with a GTR + G + I model using MEGA[54]. Sequences of CO1 used are provided in the Source Data file 1.

### Regeneration assays and foot peroxidase staining

Amputations were performed under a stereoscopic microscope using a millimetric grid to precisely measure the distance from the head. Head regeneration was measured by assessing the presence of the first two tentacle buds. Foot regeneration was primarily assessed visually by identifying basal disk morphology under a stereoscopic microscope at 144 hpa, since no new foot regeneration was observed after this point in any of the species tested. Foot regeneration data for all species shown on Fig. 1 is provided in the Source data file 2. To further test for the presence of a regenerated basal disk, foot peroxidase assays were performed in *H. oligactis* polyps nine days after amputation. Animals were relaxed in 2% urethane for 5 min, then fixed in 4% formaldehyde at room temperature for 1 h. Animals were then rinsed three times in phosphate buffer saline solution (PBS) + 0.25% Triton X-100 for 5 min each. Foot peroxidase staining was performed by incubating animals for 15 min in a solution of 0.02% diaminobenzidine, 0.25 % Triton X-100, and 0.003% hydrogen peroxide in PBS. Following staining, animals were rinsed in PBS + 0.25% Triton X-100 for 30 min, then placed in 25% glycerol for 5 min, followed by 50% glycerol for mounting.

### Lateral head organizer grafting

For each time point tested, three biological replicates of at least 10 *H. vulgaris* and 10 *H. oligactis* were bisected. A small portion of tissue from the oral wound site was excised and set aside for grafting. Host animals were punctured at the mid-section of the body using a scalpel. The regenerating tissue from the donor oral wound site was picked up with entomological tweezers and inserted into the puncture site of the host animal. The grafted tissue was held in place against the host's body column using tweezers and a dissection needle for 5 min,

allowing it to adhere. Lateral head formation in the host animals was evaluated every 24 h for 6 days, after which total number of animals with lateral heads was counted. Lateral head formation data is provided in the Source Data file 10.

### Alsterpaullone (ALP) and iCRT14 treatments

For ALP treatments, three biological replicates of at least 20 top halves and bottom halves of bisected *H. oligactis* were incubated in either 0.05% DMSO or ALP at 2.5 μM, 5 μM, or 10 μM in *Hydra* medium for 3 h. Following treatment, animals were immediately washed, and foot or head regeneration was assessed 144 h post-amputation. During this period, animals were fed 3 times per week. For iCRT14 treatments, four biological replicates of at least 29 *H. vulgaris* were incubated in 5 μM iCRT14 or 0.05% DMSO for 12 h post-bisection. After treatment, both halves of the animals were washed and monitored for head and foot regeneration every 24 h for 96 h. Animals were fed every 48 h following amputation. Foot and Head regeneration data for iCRT14 treated *H. vulgaris* are provided in Source data file 11. Foot regeneration data for ALP-treated *H. oligactis* is provided in Source data file 12. Foot regeneration data for iCRT14-treated *H. oligactis* is provided in Source data file 19.

### RNA-seq library preparation

For each regeneration sample, at least three biological replicates of 30 *H. oligactis* were starved for 48 h prior to bisection. Following 50% bisection, animals were incubated in either 0.05% DMSO or 5 μM ALP for 3 h, then washed with *Hydra* medium. For the 0 h post-amputation samples, animals were exposed to DMSO or 5 μM ALP for 30 s, then washed before RNA extraction. Regenerating animals were allowed to regenerate for 3, 12 or 48 h, after which they were rinsed with *Hydra* medium and the tissue adjacent to the oral and aboral injury sites was excised for RNA extraction. The amount of tissue collected for each sample was approximately one sixth of the total body length. For ALP-treated samples, only the aboral injury site was collected. In addition, three biological replicates of 30 animals were used to collect homeostatic head tissue by decapitating just below the tentacle ring, and three biological replicates of 30 feet (peduncle + basal disk) were obtained as homeostatic foot samples. A second batch included three biological replicates of 24 hpa in DMSO and ALP, as well as three biological replicates of 0 hpa samples in DMSO and two biological replicates of homeostatic head and foot following the same conditions. Total RNA was extracted from the excised tissue fragments using Trizol™ (Thermo Fisher Scientific), followed by DNA decontamination with DNAse I (R1013, Zymo Research). A final RNA purification step was performed using the Zymo RNA Clean and Concentrator kit (R1017, Zymo Research) according to the manufacturer's instructions. Poly(A)-enriched mRNA libraries were prepared, and 150 bp paired-end sequencing was performed on a NovaSeq 6000 by Novogene Co.

### RNA-seq and computational analysis

Low-quality base calls and sequencing adapters were filtered out using Trimmomatic V.0.36. The reference transcriptome for mapping the trimmed reads was obtained by downloading the *H. oligactis* Cold Resistant (Swiss Strain) transcriptome from https://hydratlas.unige.ch/[55]. BUSCO analysis[56], revealed a high degree of redundancy (26.3% duplicates) in this transcriptome. To reduce the redundancy, Evidentialgene tr2aacds.pl (v2017.12.21) was used to decrease redundancy[57]. BUSCO stats for both the original transcriptome and reduced transcriptome are shown in Supplementary Table 2. The resulting reduced reference (Supplementary Data file 8) was then used for mapping the trimmed reads and calculating transcript counts using RSEM[58]. For annotation, predicted coding sequences were analyzed using IntersProScan[59], and Blast against SwissProt. Also, we recovered orthogroups from a previous study in our lab, where *H. oligactis* and *H. vulgaris* predicted proteomes were analyzed against 42 other animal

proteomes[23,33], allowing the recovery of orthologs between both *Hydra* species and humans. In addition, batch effects were corrected using ComBat_seq R-package[60]. Normalization of transcript counts and differential expression analysis were performed using Edge R[61], with gene expression differences assessed using glmTreat and a false discovery rate of 0.01. Results are provided in the Source data file 4 and Source data file 13. Additionally, *H. vulgaris* counts from our previous study[19], were reanalyzed using glmTreat to calculate differentially expressed genes for comparison. Principal Component Analysis, heatmaps, and gene expression plots were generated using *in-house* scripts.

For Orthoclust analysis, FPKMs for *H. oligactis* and *H. vulgaris* mapped reads were calculated with RSEM, and reciprocal orthology was determined using ReciprocalBlastHit.py[62]. In addition, orthology was confirmed with our OrthoFinder ortholog annotations. OrthoClust was run with a *p* value of 0.001, correlation threshold of 0.975, and a Kappa value of 2.

Differences in gene expression patterns over time between DMSO and ALP treatments were analyzed using maSigPro, with Q set at 0.05 and $R^2$ at 0.8. Gene Ontology enrichment analysis of maSigPro clusters was performed using FuncAssociate 3.0.

### RNA fluorescent in situ hybridization (FISH)

FISH was performed following a previously published protocol for *H. vulgaris*[31], with the only modification being the use of 250 ng of probe per sample for hybridization. Probe detection was carried out using Alexa Fluor 594 tyramide reagent (ThermoFisher). The primers used to amplify a region of *wnt3* to synthesize the RNA probe were acquired through Integrated DNA Technologies, Inc. (*H. oligactis* FW 5′-AACTGT GGATGGCTCTTGGG-3′ and RV 5′-atttaggtgacactatagGTTGCAGTTT CCCGACGTTC-3′; *H. vulgaris*: FW 5′-GCGTTGCAGAAGGAATACGA-3′ and RV 5′- taatacgactcactatagggAACAGGTGTATTCAGGCGTCA-3′), *dlx2* primers were acquired through the same provider (*H. oligactis* FW 5′- GCCGCCATTAGACAACAACA-3′ and RV 5′- taatacgactcactatagg gGGAAATTAAACGCCCTCGCT-3′; *H. vulgaris*: FW 5′- ACGGATG-GAAGTGGAAGGGT-3′ and RV 5′- atttaggtgacactatagGTGGCCTCA-TATCGCCTGAG-3′). Confocal images were acquired using the Zeiss 980 LSM with Airyscan 2 confocal microscope. Z-stack images were analyzed with Fiji[63].

### Statistical information

Results for secondary axis formation from a regenerating graft are shown in average percentage of three different biological replicates for each time point and species. Error bars represent the standard deviation. Three biological replicates of at least 10 grafted animals were recorded for each time point and species of *Hydra* ($n_{OLI\_0hpa} = 32$, $n_{OLI\_3hpa} = 30$, $n_{OLI\_8hpa} = 33$, $n_{OLI\_16hpa} = 34$, $n_{OLI\_24hpa} = 36$, $n_{OLI\_48hpa} = 35$, $n_{AEP\_0hpa} = 31$, $n_{AEP\_3hpa} = 32$, $n_{AEP\_8hpa} = 32$, $n_{AEP\_16hpa} = 32$, $n_{AEP\_24hpa} = 35$, $n_{AEP\_48hpa} = 33$). Ectopic head frequency can be found in Source data file 10. Significance of the difference in secondary axis formation between species was evaluated through a two-way ANOVA, using the "aov" function from the "stats" package in R. This analysis showed that there is a significant difference ($p$ value = $2.55 \times 10^{-5}$, F-value = 23.91, df = 1) in secondary axis formation between *H. oligactis* and *H. vulgaris*.

Results for foot regeneration after ALP treatment with different concentrations was shown as the mean foot regeneration percentage of three biological replicates of at least 21 animals for each tested concentration; the *n* for each biological replicate were distributed as follows, DMSO: $n_1 = 26$, $n_2 = 24$, $n_3 = 24$; ALP 2.5 µM: $n_1 = 21$, $n_2 = 21$, $n_3 = 23$, $n_4 = 24$; ALP 5 µM: $n_1 = 23$, $n_2 = 23$, $n_3 = 23$; and ALP 10: $n_1 = 24$, $n_2 = 22$, $n_3 = 24$. Error bars in Fig. 4e represent standard deviation of the mean foot regeneration percentage across replicates. The difference in foot regeneration percentage between tested ALP concentrations and DMSO treated control was evaluated with a two-tailed *t*-test using the "t.test" function from the "stats" package in R. The 2.5 µM ALP showed a

significant difference when compared against DMSO controls ($p$ value = 0.0214, $t = -6.7251$, df = 2, 95 percent confidence interval: −29.049058 −6.381212, mean difference = −17.71514). The 5 µM ALP showed a significant difference when tested against DMSO controls ($p$ value = 0.01538, $t = -7.9702$, df = 2, 95 percent confidence interval: −67.42184 −20.14784, mean difference = -43.78484). Finally, 10 µM ALP showed a significant difference when compared against DMSO controls ($p$ value = 0.02139, $t = -6.7273$, df = 2, 95 percent confidence interval: −48.14650 −10.58366, mean difference = −29.36508). Experimental details about foot regeneration in ALP-treated *H. oligactis* are provided in the Source data file 12.

Results for head and foot regeneration for iCRT14-treated *H. vulgaris* are shown as a mean percentage of three different biological replicates distributed as follows, head regeneration in iCRT14-treated animals: $n_1 = 29$, $n_2 = 30$, $n_3 = 30$; and DMSO control animals: $n_1 = 29$, $n_2 = 29$, $n_3 = 29$. Foot regeneration in iCRT14-treated animals: $n = 30$, $n_2 = 30$, $n_3 = 30$; and DMSO controls: $n_1 = 29$, $n_2 = 30$, $n_3 = 30$. Error bars in Fig. 4b, c represent the standard deviation in the mean foot and head regeneration percentages observed across replicates. Experimental details for iCRT14-treated *H. vulgaris* can be found in Source data file 11. Significance of the difference in foot regeneration and head regeneration percentage across time between conditions in *H. vulgaris* was evaluated through a two-way ANOVA using the "aov" function from the "stats" package in R. This analysis showed that there is a significant difference in foot regeneration percentage (two-way ANOVA, $p$ value = 0.002824, F-value = 4.99, df = 12) between DMSO-treated and iCRT14-treated *H. vulgaris*. No statistical difference was found in head regeneration percentage between DMSO-treated and iCRT14 treated *H. vulgaris*.

### Reporting summary

Further information on research design is available in the Nature Portfolio Reporting Summary linked to this article.

## Data availability

The data generated in this study are provided in the Supplementary Information and Source Data files. Source data are provided with this paper. FASTQ files for raw RNA-seq reads and raw counts from RNA-seq are deposited in the Gene Expression Omnibus under the Bio Project accession number PRJNA1231128. Data generated in this study are also available through Figshare under: https://doi.org/10.6084/m9.figshare.28620101 Source data are provided with this paper.

## Code availability

All scripts for data analysis used in this study are available through GitHub under: https://github.com/cejuliano/oligactis_foot_regeneration. This GitHub release version can be cited using https://doi.org/10.5281/zenodo.17476370.

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

## Acknowledgements

We thank Bert Hobmayer and Jack F. Cazet for their constructive comments on the manuscript and all members of Juliano Lab for discussion and suggestions during the development of this work. We also thank Bert Hobmayer for providing the *Hydra oligactis* Innsbruck 12 strain and Brigitte Galliot for providing the *Hydra oligactis* Swiss strain. This work was supported by a National Institutes of Health (NIH) grant R35 GM133689 (to C.E.J.), and a Human Frontiers Science Program Postdoctoral Fellowship (Reference number: LT000496/2020-L; to S.E.C).

## Author contributions

C.E.J. and S.E.C. conceptualized and designed the study. S.E.C., S.N., J.C., J.T., B.D.C., and C.C. performed experiments. S.E.C. performed data analysis. C.E.J. oversaw all the experiments. R.B.S., D.E.M., and R.E.S. collected and identified *Hydra* strains. S.E.C. and C.E.J. contributed to writing of the original draft, and the remaining authors read and critically revised the manuscript.

## Competing interests

The authors declare no competing interests.
