## [Peer Review file · Nature Communications]

Wnt signaling restores evolutionary loss of robust foot regeneration rates in Hydra

Corresponding Author: Dr Celina Juliano

Version 0:

Reviewer comments:

Reviewer #1

(Remarks to the Author)

See attachment.

Reviewer #2

(Remarks to the Author)

In this manuscript, the authors investigate the differential regeneration capabilities of two species within the genus Hydra. Using time-course RNA-seq analysis and comparative transcriptomic approaches between *H.vulgaris* and *H.oligactis*, the authors shed light on mechanisms that correspond with impaired aboral regeneration abilities in *H.oligactis*. Using pharmacological approaches targeting the Wnt pathway, they show that regeneration abilities can be partially restored in *H. oligactis* by modulating the Wnt pathway.

Similar comparative studies of regeneration capabilities have been done in planarians, having revealed similar involvement of the Wnt pathway in the loss of regenerative potential in closely related flatworm species. This manuscript shows that changes in Wnt signaling may be involved in modulating regeneration across 650 million years of evolution spanning the cnidarian-planarian divergence. This paper is therefore important for addressing a major question in the field: how regenerative mechanisms have been altered to shapes regeneration abilities across metazoan evolution. Overall, the evidence provided by the authors supports their claims. However, some more data are need to enhance the rigor of the work and some rewriting is needed to better explain and contextualize the advance made by this study.

Presenting planarian literature at the outset

Given that previous work (cited below) in planarians highlights the role of Wnt signaling in governing regeneration capabilities across closely related species, the authors should mention these studies in their introduction to establish the importance of their work. This manuscript is showing that such phenomena are not limited to planarians and point to a broader role for Wnt signaling in regulating regenerative potential across animal phyla. Replacing the generic statement in the introduction about how “very little is known”, explaining that this work has been done in another phylum actually elevates this paper, placing it as an important contribution to a field of researchers to whom the question of differential regenerative abilities is a major pursuit.

1. Sikes, J., Newmark, P. Restoration of anterior regeneration in a planarian with limited regenerative ability. *Nature* 500, 77–80 (2013)
2. Liu, SY., Selck, C., Friedrich, B. et al. Reactivating head regrowth in a regeneration-deficient planarian species. *Nature* 500, 81–84 (2013).
3. Vila-Farré, M., Rozanski, A., Ivanković, M. et al. Evolutionary dynamics of whole-body regeneration across planarian flatworms. *Nat Ecol Evol* 7, 2108–2124 (2023)

Deeper mechanistic investigation

Taking planarian and Hydra studies together, it is striking that Wnt pathway components operate with similar dynamics and

possibly with similar functions – e.g. Wnt ligand mRNA is upregulated at both wound sites at first but later abolished on one side (aboral in Hydra and anterior in planarians). In both cases, a high Wnt environment corresponds to specific tissue identities - tail identity in planarians and oral identity in Hydra. Studies in regeneration-deficient planarians, where tail fragments are unable to regenerate heads, have shown that this defect is due to sustained Wnt signaling. Downregulation of Wnt in those contexts' rescues head regeneration. Given this, it is important to establish what happens if the authors downregulate Wnt pathway in *H. oligactis*. Would it also rescue the foot regeneration defect? This study emphasizes the role of injury-induced Wnt signaling in regulating foot regeneration. However, expanding the scope to consider Wnt's broader involvement in defining oral and aboral identities could help contextualize these findings in a broader evolutionary framework. For instance, treating foot-less *H. oligactis* polyps with a Wnt antagonist would be an important experiment to do.

Use of more accurate terms – loss vs. spatial differences in regenerative response

Based on previous studies and this one, it is evident that *H. oligactis* hasn't completely lost its regeneration ability. It regenerates its oral end just fine, and the probability of aboral regeneration can be very high depending on where the amputation plane is. Further, even at the amputation plane chosen in this paper, some of the animals still are able to regenerate just fine. Therefore, the use of the word "loss" in the title is misleading, as this trait still exists in the population. The authors should replace the term loss with something that capture the phenomenon more accurately.

Additionally, in the introduction (line 68), the authors claim to have "discovered that foot regeneration ability has been lost in the *Oligactis* clade of the Hydra genus, including lab species *Hydra oligactis*." However, from Hoffmeister (1991), it is already known that *H. oligactis* has impaired foot regeneration capabilities. The authors should rewrite this line to be true to what was known before.

Reconciling current results with previously published data

Comparing the results from the Hoffmeister (1991) study: why does the frequency of foot regeneration in *H. oligactis* differ from this study? In Hoffmeister (1991), it was reported that when amputated in half, 24% of *H. oligactis* regenerated feet by 50 hpa, while in this study, it is around 10% even at 144 hpa. The authors should address this discrepancy in foot regeneration frequency between the two studies. Is it due to different amputation planes, strains, age, nutrient conditions, or other factors?

Absence vs delay in regeneration

The authors should mention the rationale for choosing the temporal window to assess regeneration dynamics of *H. oligactis*. How was the 144 hpa endpoint chosen? Is it possible that the animal would eventually regenerate given a longer time window? Given that *H. oligactis* is about twice the size of *H. vulgaris*, could this result in delayed foot regeneration in *H. oligactis*?

Rigorous orthology assignment

The orthology analysis between the two Hydra species is based solely on reciprocal BLAST, which can sometimes lead to false ortholog detection. This approach is appropriate for genome/transcriptome-wide comparisons, but a more precise approach is needed when specific genes become the focus of a study. If this has not been previously published, the authors should incorporate a phylogenetic framework for at least the candidate genes *wnt3*, *wnt7*, *wnt9/10c*, and *brachyury* to strengthen their claim regarding comparative expression dynamics between *H. vulgaris* and *H. oligactis*.

Wound response in wild type animals

Given that ALP treatment leads to downregulation of wound-induced bZIP TFs (*creb*, *jun*, *fos*) at 3 hpa, the authors should provide data on what happens to these bZIP TFs under normal conditions in *H. oligactis*. In lines 449–450, the authors mention comparing bZIP TFs between *H. oligactis* and *H. vulgaris*, but the associated graph in Fig. S8 only shows ALP and DMSO treatment within *H. oligactis*. The authors should include comparative data between *H. vulgaris* and *H. oligactis*, as it is important to establish whether general wound response factors are differentially upregulated between the two species and even at different amputation planes in *H. oligactis*. These data are essential for fully making sense of the Wnt data.

Rigor in drug treatments

For their pharmacological treatment, the authors used only one drug—Alsterpullone (ALP)—as a Wnt signaling activator in *H. oligactis*. Even though ALP has been previously applied in *H. vulgaris*, its specificity in *H. oligactis* needs to be tested. The authors should also use more than one Wnt agonist to establish Wnt-specific roles in regeneration in *H. oligactis*. Additionally, most of the experiments rely on RNA sequencing to validate gene expression under different conditions. However, since *H. oligactis* is amenable to Fluorescence In-Situ Hybridization (FISH), adding FISH experiments to show the spatial location of Wnt signaling activation during foot regeneration would help strengthen the claim that Wnt signaling is involved.

Lastly, what happens to head regeneration under ALP treatment? Does it affect head regeneration dynamics?

Minor Comments:

- Line 87: The citation for the high-quality genome does not match the paper cited, which is about *H. vulgaris*. I could not find any genomic information for *H. oligactis*—please double-check.
- In Fig. 5C,D: The color difference between "foot-specific ALP 5 μ M" and "non-specific" is not very distinct and obscures the contrast. Using a more contrasting red/blue color scheme may help.
- Citation in line 437–438 appears incomplete.

Reviewer #3

(Remarks to the Author)

Reviewer #4

(Remarks to the Author)

This manuscript represents a comprehensive comparative analysis of foot regeneration ability between two species of Hydra – *Hydra vulgaris* where the trait is present, and *Hydra oligactis* where the trait is absent. It has been lost in the entire *oligactis* clade. The goal of the study is to uncover the mechanisms required to initiate foot regeneration in Hydra. Bulk RNAseq analysis over a time course of regeneration was performed for *H. oligactis* and compared with previous results published for *H. vulgaris*. The major finding was that Wnt signalling activation is significantly lower in *H. oligactis* upon injury. Pharmaceutical manipulation with two different drugs demonstrated that foot regeneration can be rescued with Wnt pathway activation. A focus on the transcriptional regulators involved in foot regeneration resulted in the identification of *dlx2* as an important, early transcription factor that is foot specific.

Overall, the manuscript is clearly written, the authors do a good job of describing carefully executed experiments and analysis, and they tell a clear story. I found all aspects of experimental design including the bulk tissue RNA-seq sampling, the analyses described, and the figures supporting the results to be straightforward and logical. I also appreciate the availability of the data provided.

The major issue I identified to be addressed is in the framing of the manuscript, including placing the current study in a larger context of established literature both in the Introduction and the Discussion. The authors do not do a good job of framing the study based on previous work or adequately putting these results into a larger context with similar studies in other animals which also focused on the Wnt/B-catenin signaling pathway related to regeneration gain and loss. Even the *dlx2* transcription factor had been previously shown to inhibit foot regeneration in *H. vulgaris* so finding it again was not particularly surprising. The fact that the result was confirmatory was only mentioned briefly in the discussion.

This paper immediately calls to mind a trio of papers that came out together in Nature in 2013 that compared closely related flatworm species with differing injury and regeneration responses – Liu et al. 2013, Sikes et al. 2013, and Umesono et al. 2013. The focus of the three papers was on head regeneration from tail fragments in regeneration-deficient versus regeneration-competent species and all three papers converge on the Wnt/B-catenin pathway and that reducing or downregulating the activity of this pathway leads to head regeneration in tail fragments that would not normally regenerate. More recently, Vila-Farré et al. 2023 in Nature Ecology and Evolution did a broader sampling of 40 planarian species to examine the evolution of head regeneration in this group of animals. One of their broad findings was that the Wnt/B-catenin pathway is linked to the emergence of planarian regeneration defects.

Although the details in the present manuscript are not precisely the same as in these studies and is focused on species of the cnidarian Hydra and on foot regeneration rather than head regeneration, the inspiration for this work and the framework for looking into what underlies differences regenerative potential in closely related species within a particular taxonomic group must surely come from these well-known papers which had a big impact when they were published. As such, I was very surprised to see that only one of these papers was cited (Sikes et al. 2013) and that it was only briefly mentioned in the Discussion.

The authors missed an opportunity to review these breakthrough studies of loss of regenerative ability in closely related species of flatworms and the implication of the Wnt/B-catenin pathway in modulating this ability. The overall approach taken in the present manuscript is also reminiscent of these studies, so I feel that they must be mentioned and briefly reviewed early in the paper, in the introduction, with a few clarifying statements about how studying this same general phenomenon in closely related species of Hydra in the context of foot regeneration centered on the same signaling pathway provides new insight. The way it is currently phrased in this manuscript's introduction, it sounds like there has been little progress in understanding of mechanisms driving loss of regenerative potential in any animal group, which is simply not the case. Giving more context based on these studies and their findings will both enhance the present manuscript and give a better framework for focusing the current study.

This is the current text that needs to be substantially rewritten to reflect the suggestions above:

(lines 24-30)

"Many studies have already advanced our understanding of the genes and pathways that control regeneration in highly regenerative animals 3,4. However, we have little understanding of the mechanisms that drive loss of regenerative potential. Comparative approaches between closely related organisms with contrasting regeneration abilities offer a promising avenue to both reveal the mechanisms of regeneration, as well as uncover how regeneration has been

shaped throughout evolution.”

In the Discussion, these same 2013 and 2023 planarian studies should be discussed again in the context of the results of the present study.

The current Discussion related to this is limited to a few sentences near the end of the Discussion and the focus is on differences in reproductive strategies between species in the *Oligactis* group of *Hydra* versus other groups: (lines 503-511)

“One possibility is that the foot regeneration defect in *H. oligactis* evolved as a trade-off associated with semelparous reproduction. To explore this possibility, future research should explore the potential function of Wnt signaling in the induction of gametogenesis in

H. oligactis. Similar trade-offs involving Wnt signaling may occur in other animals.

For example, the semelparous planarian worm *Proctotyla fluviatilis* cannot regenerate its head when cut near the tail due to excessive Wnt signaling activation upon injury; pharmacological downregulation of Wnt signaling rescues regeneration 44.”

This idea of different reproductive strategies having trade-offs with regenerative ability has been raised before in the literature and is also directly addressed in the Vila-Farré et al. 2023 paper (from the abstract):

“Our finding that Wnt signalling has multiple roles in the reproductive system of the model species *Schmidtea mediterranea* raises the possibility that a trade-off between egg-laying, asexual reproduction by fission/regeneration and Wnt signalling drives regenerative trait evolution. Although quantitative comparisons of Wnt signalling levels, yolk content and reproductive strategy across our species collection remained inconclusive, they revealed divergent Wnt signalling roles in the reproductive system of planarians.”

In summary, I suggest that the authors work to reframe and provide more context for this study by adding these papers to the introduction and the discussion to acknowledge that these ideas have been previously raised and explored, at least in a broader sense, and a similar approach was taken to identify what factors underline the differences in regenerative ability in a different group of animals and that this effort converged on identifying the Wnt/B-catenin pathway as key to these questions. Without this proper context, the reader is left with an incorrect impression of the novelty of the approach and the results from the present work.

Beyond this inadequate framing and lack of proper context, the study itself is solid and provides a good basis and foundation for future study by setting up a new detailed comparison between two cnidarian species with differing abilities to regenerate their foot.

Version 1:

Reviewer comments:

Reviewer #2

(Remarks to the Author)

In the manuscript “Wnt signaling restores evolutionary loss of robust foot regeneration rates in *Hydra*,” the authors examine regenerative capabilities in the genus *Hydra* and explore the molecular basis of foot regeneration abilities in *Hydra oligactis*. Using regeneration time-course RNA-seq analysis and a comparative transcriptomics approach between *Hydra vulgaris* and *Hydra oligactis*, they reveal that a reduced transcriptional program, including the attenuation of the injury-induced Wnt pathway, may be linked to the decreased foot regeneration rates in *Hydra oligactis*. This finding highlights the role of Wnt signaling in governing regenerative abilities and dynamics across animals (discussed thoroughly in the discussion section of the manuscript). This and similar comparative studies from planarians have laid the groundwork for investigating the evolutionary mechanisms underlying regeneration abilities in animals. Specifically, these studies have established a foundation for exploring how Wnt signaling regulation and its dynamics relate to regenerative capabilities among different species.

The authors have done a great job of incorporating feedback from the earlier reviews and, as a result, performed new experiments (assessing foot regeneration with Wnt inhibition), elaborated on their findings, and better contextualized their results (comparing and contrasting results from planarian studies). This exercise has reinforced the importance of their work and built a refined conceptual understanding in this field.

Finally, this study brings new insights to the field, and the data presented here support their claims, which makes this manuscript worth publishing in *Nature Communications*.

Reviewer #3

(Remarks to the Author)

Reviewer #4

(Remarks to the Author)

I am very pleased with the revision of this manuscript and the response to the reviewers. I feel that the authors have adequately addressed my concerns which mostly centered on the framing of the manuscript and putting it into context with previous literature. I am happy with how the manuscript is now framed and presented and the references which are now properly cited. It is now much clearer which are the novel aspects of this study in relation to what was previously known. I believe this paper will make a strong contribution to the field and I appreciate how thorough the authors were with providing the underlying data for their study.

Response to Reviewers

Reviewer 1

Campos et al. examine the genetic basis of why some hydra species fail to regenerate their foot region. They compared a species of Hydra, *Hydra oligactis*, that frequently does not regenerate their foot to regeneration of *Hydra vulgaris*. Through analysis of single cell sequencing of regenerating and non-regenerating Hydra species, they found that the wnt genes were some of the genes notably not being upregulated in *H. oligactis*. Remarkably, when the researchers stimulated wnt signaling with an agonist, they found the ability to regenerate the foot structures were restored. Conversely, when wnt signaling was inhibited in *H. vulgaris*, foot regeneration was delayed.

Regeneration is an interesting, but poorly understood process. It is especially unclear how and why some species regenerate robustly, and other species may fail to regenerate, or, in the case of *H. oligactis*, take longer. This approach of comparing Hydra species that exhibit different regeneration capacity allow some insight into how change in expression of early genes results in the loss of the ability to regenerate. This is exciting work that will be of interest to a large number of readers.

The manuscript is beautifully written, and the data are presented well. The methods are detailed and would allow for this approach to be used on other systems where there are closely related species that have different regenerative capacity. There is only one confusing part that could be clarified or presented differently.

We thank the reviewer for their positive assessment of our manuscript and we have addressed the comments below.

The graphs in Figure 3, showing the different levels of wnt expression in the two species have different axes on them, making the direct comparison of them confusing.

Thank you for this suggestion, we have changed this in Figure 3 and agree that this is easier for direct comparisons.

Also, the double headed regenerates shown in Supplemental Figure 5 is spectacular! I don't know if this phenotype has been reported before in hydra before, but it seem that it would merit a paragraph in the discussion, at least.

We have expanded the Discussion (Lines 492–496) to note that successful foot regeneration at the aboral end requires a precise balance, one that avoids tipping toward head regeneration. We now also cite the double-head phenotype previously reported for *H. vulgaris* as another example of this balance being disrupted.

Reviewer 2

In this manuscript, the authors investigate the differential regeneration capabilities of two species within the genus Hydra. Using time-course RNA-seq analysis and comparative transcriptomic approaches between *H. vulgaris* and *H. oligactis*, the authors shed light on mechanisms that correspond with impaired aboral regeneration abilities in *H. oligactis*. Using pharmacological approaches targeting the Wnt pathway, they show that regeneration abilities can be partially restored in *H. oligactis* by modulating the Wnt pathway.

Similar comparative studies of regeneration capabilities have been done in planarians, having revealed similar involvement of the Wnt pathway in the loss of regenerative potential in closely related flatworm species. This

manuscript shows that changes in Wnt signaling may be involved in modulating regeneration across 650 million years of evolution spanning the cnidarian-planarian divergence. This paper is therefore important for addressing a major question in the field: how regenerative mechanisms have been altered to shapes regeneration abilities across metazoan evolution. Overall, the evidence provided by the authors supports their claims. However, some more data are need to enhance the rigor of the work and some rewriting is needed to better explain and contextualize the advance made by this study.

We thank the reviewer for their thoughtful evaluation of our manuscript. In response, we have substantially improved the framing, refined the language for greater precision, and added additional supporting data. Altogether, we believe these revisions have strengthened the manuscript considerably. Please find our point-by-point responses below.

Presenting planarian literature at the outset

Given that previous work (cited below) in planarians highlights the role of Wnt signaling in governing regeneration capabilities across closely related species, the authors should mention these studies in their introduction to establish the importance of their work. This manuscript is showing that such phenomena are not limited to planarians and point to a broader role for Wnt signaling in regulating regenerative potential across animal phyla. Replacing the generic statement in the introduction about how “very little is known”, explaining that this work has been done in another phylum actually elevates this paper, placing it as an important contribution to a field of researchers to whom the question of differential regenerative abilities is a major pursuit.

1. Sikes, J., Newmark, P. Restoration of anterior regeneration in a planarian with limited regenerative ability. *Nature* 500, 77–80 (2013)
2. Liu, SY., Selck, C., Friedrich, B. et al. Reactivating head regrowth in a regeneration-deficient planarian species. *Nature* 500, 81–84 (2013).
3. Vila-Farré, M., Rozanski, A., Ivanković, M. et al. Evolutionary dynamics of whole-body regeneration across planarian flatworms. *Nat Ecol Evol* 7, 2108–2124 (2023)

We thank the reviewer for noting that we did not properly frame our findings within the broader context of the literature at the outset of the paper. In response, we have substantially revised the Introduction to include a discussion of the extensive planarian literature in the opening paragraph, as well as the concept that modulations in Wnt signaling may underlie major evolutionary shifts in regenerative outcomes (Lines 28-35). We then highlight the need for additional comparative platforms (Lines 35-41), and in the next paragraph, describe the specific advantages of comparisons within the cnidarian phylum, which occupies a key phylogenetic position as the sister group to bilaterians (Lines 42-45). We have also included a much deeper discussion of the topics in the Discussion section (Lines 472-498). We thank the reviewer for prompting these changes, which have substantially improved the framing of the manuscript and better placed our work within the broader context of the existing literature.

Deeper mechanistic investigation

Taking planarian and Hydra studies together, it is striking that Wnt pathway components operate with similar dynamics and possibly with similar functions – e.g. Wnt ligand mRNA is upregulated at both wound sites at first but later abolished on one side (aboral in Hydra and anterior in planarians). In both cases, a high Wnt environment corresponds to specific tissue identities - tail identity in planarians and oral identity in Hydra. Studies in regeneration-deficient planarians, where tail fragments are unable to regenerate heads, have shown that this defect is due to sustained Wnt signaling. Downregulation of Wnt in those contexts’ rescues head regeneration. Given this, it is important to establish what happens if the authors downregulate Wnt pathway in *H. oligactis*. Would it also rescue the foot regeneration defect? This study emphasizes the role of

injury-induced Wnt signaling in regulating foot regeneration. However, expanding the scope to consider Wnt's broader involvement in defining oral and aboral identities could help contextualize these findings in a broader evolutionary framework. For instance, treating foot-less *H. oligactis* polyps with a Wnt antagonist would be an important experiment to do.

We thank the reviewer for this suggestion, which has prompted us to do a better job contextualizing our findings on the role of Wnt signaling in *Hydra* regeneration. As noted by the reviewer, in planarians, decreasing Wnt signaling can rescue head regeneration defects caused by *excessive* anterior Wnt activity. In contrast, our data suggest that in *H. oligactis*, the reduced foot regeneration rate is due to *insufficient* injury-induced Wnt signaling at the aboral end. While lack of Wnt signaling is ultimately required for foot regeneration in *Hydra*, an early transient burst is necessary for high regeneration rates, a role distinct from axial patterning.

To test this idea further, we took the reviewer's suggestion, and treated *H. oligactis* with a Wnt antagonist. We found that this treatment slightly reduced the already low foot regeneration rate, rather than increasing it as in planarians (See new Fig. S9, described in lines 389-395). This supports our conclusion that Wnt signaling in *Hydra* has an injury-specific role in foot regeneration, distinct from its axial patterning function.

Use of more accurate terms – loss vs. spatial differences in regenerative response

Based on previous studies and this one, it is evident that *H. oligactis* hasn't completely lost its regeneration ability. It regenerates its oral end just fine, and the probability of aboral regeneration can be very high depending on where the amputation plane is. Further, even at the amputation plane chosen in this paper, some of the animals still are able to regenerate just fine. Therefore, the use of the word "loss" in the title is misleading, as this trait still exists in the population. The authors should replace the term loss with something that capture the phenomenon more accurately.

We agree with the reviewer and have revised the title and manuscript throughout to make the language more precise. We agree that it is incorrect to state that *H. oligactis* has lost regeneration "potential." What we actually observe is a context-dependent reduction in the rate of foot regeneration. We have updated the title to (1) specify the spatial aspect by explicitly referring to foot regeneration, and (2) replace "loss of regenerative potential" with "loss of robust foot regeneration rates."

We retained the term "loss" because it reflects the evolutionary trajectory of this specific trait in *Hydra*. As shown in Figure 1, reduced foot regeneration rates occur in a specific clade, suggesting an evolutionary loss of high foot regeneration rates rather than a gain in other species. We have made similar changes throughout the text, too many to list individually, to replace the concept of "reduced regenerative potential" with the concept of "reduced rate of regeneration."

As one example, in the Discussion we changed (Lines 571-573):

"Our findings suggest that foot regeneration ability is ancestral in the *Hydra* genus but was lost in the *Oligactis* group."

to:

"Our findings suggest that robust foot regeneration ability is ancestral in the *Hydra* genus and that reduced foot regeneration rates are a derived trait of the *Oligactis* group."

Additionally, in the introduction (line 68), the authors claim to have "discovered that foot regeneration ability has been lost in the *Oligactis* clade of the *Hydra* genus, including lab species *Hydra oligactis*." However, from Hoffmeister (1991), it is already known that *H. oligactis* has impaired foot regeneration capabilities. The authors should rewrite this line to be true to what was known before.

We have clarified in Lines 53–57, early in the Introduction, that the *H. oligactis* foot regeneration deficit had been reported prior to our work. In Lines 77–79, we state that our findings “confirm” this previously described deficit and extend it to the *Oligactis* clade, thereby clarifying the evolutionary history of foot regeneration phenotypes within the genus.

Reconciling current results with previously published data

Comparing the results from the Hoffmeister (1991) study: why does the frequency of foot regeneration in *H. oligactis* differ from this study? In Hoffmeister (1991), it was reported that when amputated in half, 24% of *H. oligactis* regenerated feet by 50 hpa, while in this study, it is around 10% even at 144 hpa. The authors should address this discrepancy in foot regeneration frequency between the two studies. Is it due to different amputation planes, strains, age, nutrient conditions, or other factors?

We have added text (Lines 127–135) to address this discrepancy. Although we cannot determine the exact cause, we tested two strains of *H. oligactis* in our lab and obtained identical results. This suggests that culture conditions may be an important factor.

Absence vs delay in regeneration

The authors should mention the rationale for choosing the temporal window to assess regeneration dynamics of *H. oligactis*. How was the 144 hpa endpoint chosen? Is it possible that the animal would eventually regenerate given a longer time window? Given that *H. oligactis* is about twice the size of *H. vulgaris*, could this result in delayed foot regeneration in *H. oligactis*?

We have now clarified in the text (Lines 103-107) that in our initial experiments, we monitored animals for up to two months and feet never returned. In these long term experiments, we never saw foot regeneration after 5 days (120 hours), so we conservatively established 6 days (144 hours) as the endpoint of our assays for the remainder of our study.

Rigorous orthology assignment

The orthology analysis between the two Hydra species is based solely on reciprocal BLAST, which can sometimes lead to false ortholog detection. This approach is appropriate for genome/transcriptome-wide comparisons, but a more precise approach is needed when specific genes become the focus of a study. If this has not been previously published, the authors should incorporate a phylogenetic framework for at least the candidate genes *wnt3*, *wnt7*, *wnt9/10c*, and *brachyury* to strengthen their claim regarding comparative expression dynamics between *H. vulgaris* and *H. oligactis*.

We agree with the reviewer on the importance of having a robust analysis to establish direct orthology for the genes mentioned in our manuscript between *H. oligactis* and *H. vulgaris*. We apologize for not describing our methodology more clearly. In our OrthoClust analysis, gene orthology was initially assigned using reciprocal BLAST with the transcriptomic references of both species. We addressed the limitations of this strategy by annotating gene orthology and functions with OrthoFinder, using both *Hydra* species and a human reference transcriptome. Additionally, we verified gene orthology by comparing our results to the orthogroups reported in Cazet et al., 2023, which analyzed 44 metazoan proteomes, including predicted transcripts for *H. oligactis* and *H. vulgaris*. Importantly, OrthoFinder infers orthologs by building gene trees, making the inference phylogenetically based. We updated the text (Lines 250-257) to more accurately describe and reference our methodology.

Wound response in wild type animals

Given that ALP treatment leads to downregulation of wound-induced bZIP TFs (*creb*, *jun*, *fos*) at 3 hpa, the authors should provide data on what happens to these bZIP TFs under normal conditions in *H. oligactis*. In lines

449–450, the authors mention comparing bZIP TFs between *H. oligactis* and *H. vulgaris*, but the associated graph in Fig. S8 only shows ALP and DMSO treatment within *H. oligactis*. The authors should include comparative data between *H. vulgaris* and *H. oligactis*, as it is important to establish whether general wound response factors are differentially upregulated between the two species and even at different amputation planes in *H. oligactis*. These data are essential for fully making sense of the Wnt data.

We have added the *H. vulgaris* data to Figure S11 (formerly Figure S9) to allow the reader to make the direct comparison of expression levels between the two species.

Rigor in drug treatments

For their pharmacological treatment, the authors used only one drug—Alsterpullone (ALP)—as a Wnt signaling activator in *H. oligactis*. Even though ALP has been previously applied in *H. vulgaris*, its specificity in *H. oligactis* needs to be tested. The authors should also use more than one Wnt agonist to establish Wnt-specific roles in regeneration in *H. oligactis*.

We appreciate the reviewer's thoughtful comment regarding the use of pharmacological reagents for mechanistic studies in a new model organism. We agree that caution is warranted, and we have taken several steps to ensure the reliability of our conclusions. First, we believe that we have adequately validated the activity of Alsterpullone in *Hydra oligactis*. Specifically, (1) RNA-seq data demonstrate strong transcriptional upregulation of canonical Wnt pathway genes, consistent with the known positive feedback and self-amplification dynamics of this pathway (e.g. Fig. S10); and (2) treatment of uninjured *H. oligactis* produces the same ectopic tentacle phenotype observed in *Hydra vulgaris*, consistent with Wnt pathway activation; we have now included these results as a new supplemental figure in the revised manuscript (Supp Fig. 5, Lines 329-335). In addition, to complement our activation studies, we treated *H. vulgaris* with a previously validated Wnt inhibitor (see Gufler et al., 2018; Cazet et al., 2021). This treatment phenocopied the foot regeneration defect observed in *H. oligactis* (Supp Fig. 4B, Fig. S7), providing *independent evidence* that modulation of Wnt signaling influences regeneration outcomes in both species. Together, these results, obtained using two pharmacological reagents with established activity, support the central conclusion that Wnt pathway activity regulates foot regeneration outcomes in *Hydra*. While we agree that testing additional Wnt agonists would further strengthen the mechanistic framework, we believe that the current data already provide a compelling case. Moreover, rigorously validating a second agonist would require extensive additional work, including phenotypic and transcriptomic validation, as well as careful titration of dose and timing to modulate Wnt signaling in a manner that shifts regeneration specifically toward foot identity rather than head induction. Given the strength of the current evidence, we respectfully argue that this would represent a disproportionate burden for the scope of this study.

Additionally, most of the experiments rely on RNA sequencing to validate gene expression under different conditions. However, since *H. oligactis* is amenable to Fluorescence In-Situ Hybridization (FISH), adding FISH experiments to show the spatial location of Wnt signaling activation during foot regeneration would help strengthen the claim that Wnt signaling is involved.

We have attempted this approach, but the FISH assay was not sensitive enough. We encountered a similar limitation in our previous work in *H. vulgaris* (Cazet et al., 2021), where we were unable to detect the transient aboral Wnt pulse by FISH, despite its clear detection by RNA-seq. RNA-seq remains the most quantitative method for this purpose, and in all of our regeneration experiments, RNA is extracted specifically from regenerating tips rather than whole animals, providing some spatial resolution.

Importantly, in our *H. oligactis* experiments, the entire fragment is treated with ALP, so a localized activation is not expected. This comment made us realize that our diagrams in Figure 4 were misleading, therefore we have updated these to better reflect the drug treatment.

Lastly, what happens to head regeneration under ALP treatment? Does it affect head regeneration dynamics?

We thank the reviewer for suggesting this experiment, which has led to the addition of a new figure (Fig. S6) and interesting new results (Lines 336–346). We found that head regeneration in *H. oligactis* increased in response to ALP treatment, consistent with our overall model and further supporting the validity of ALP for activating Wnt signaling in *H. oligactis*.

Minor Comments:

- Line 87: The citation for the high-quality genome does not match the paper cited, which is about *H. vulgaris*. I could not find any genomic information for *H. oligactis*—please double-check.

We missed a citation here, thank you for noticing, we fixed this. (Now line 98)

- In Fig. 5C,D: The color difference between “foot-specific ALP 5 μ M” and “non-specific” is not very distinct and obscures the contrast. Using a more contrasting red/blue color scheme may help.

The reviewer was correct that this color combination was not suitable for color-blind readers, as teal and gray appear similar. We have therefore updated the coloring in Figures 5, S10, S11, and S12 to a palette that is color-blind friendly.

- Citation in line 437–438 appears incomplete.

Thank you for noticing this omission. We have now added the missing references (now Line 508)

Reviewers 3 and 4

This manuscript represents a comprehensive comparative analysis of foot regeneration ability between two species of Hydra – *Hydra vulgaris* where the trait is present, and *Hydra oligactis* where the trait is absent. It has been lost in the entire *oligactis* clade. The goal of the study is to uncover the mechanisms required to initiate foot regeneration in Hydra. Bulk RNAseq analysis over a time course of regeneration was performed for *H. oligactis* and compared with previous results published for *H. vulgaris*. The major finding was that Wnt signalling activation is significantly lower in *H. oligactis* upon injury. Pharmaceutical manipulation with two different drugs demonstrated that foot regeneration can be rescued with Wnt pathway activation. A focus on the transcriptional regulators involved in foot regeneration resulted in the identification of *dlx2* as an important, early transcription factor that is foot specific.

Overall, the manuscript is clearly written, the authors do a good job of describing carefully executed experiments and analysis, and they tell a clear story. I found all aspects of experimental design including the bulk tissue RNA-seq sampling, the analyses described, and the figures supporting the results to be straightforward and logical. I also appreciate the availability of the data provided.

We thank the reviewer for their positive overall assessment and for providing valuable feedback on the framing of our manuscript. We have made substantial updates to address these points. Please see our point-by-point discussion below.

The major issue I identified to be addressed is in the framing of the manuscript, including placing the current study in a larger context of established literature both in the Introduction and the Discussion. The authors do not do a good job of framing the study based on previous work or adequately putting these results into a larger context with similar studies in other animals which also focused on the Wnt/B-catenin signaling pathway related to regeneration gain and loss. Even the *dlx2* transcription factor had been previously shown to inhibit foot regeneration in *H. vulgaris* so finding it again was not particularly surprising. The fact that the result was confirmatory was only mentioned briefly in the discussion.

We fully agree with the reviewer that the manuscript needed stronger framing within the broader context. To address this, we made substantial additions to both the Introduction (Lines 28-41) and the Discussion (Lines 472-498). Please also see our response to Reviewer 2 for more details.

Regarding *dlx2*, in the Results section (Lines 447-449) we have better emphasized the findings of the previous study. In the Discussion (Lines 549-558), we expanded our discussion of *dlx2* to highlight this earlier work and explain how our study extends it. Specifically, our study provides evidence that *dlx2* functions as a high-level regulator, being the first TF activated during rescued *H. oligactis* foot regeneration, and we demonstrate that its expression is Wnt-dependent. To underscore this point, we revised panel H of Figure 5: instead of showing the *dlx2* expression pattern alone, we now present heatmaps of fold-change expression for foot-specific TFs under control and ALP-treated conditions. These data show that *dlx2* is the earliest TF upregulated by pharmacological Wnt activation.

This paper immediately calls to mind a trio of papers that came out together in Nature in 2013 that compared closely related flatworm species with differing injury and regeneration responses – Liu et al. 2013, Sikes et al. 2013, and Umesono et al. 2013. The focus of the three papers was on head regeneration from tail fragments in regeneration-deficient versus regeneration-competent species and all three papers converge on the Wnt/B-catenin pathway and that reducing or downregulating the activity of this pathway leads to head regeneration in tail fragments that would not normally regenerate. More recently, Vila-Farré et al. 2023 in Nature Ecology and Evolution did a broader sampling of 40 planarian species to examine the evolution of head regeneration in this group of animals. One of their broad findings was that the Wnt/B-catenin pathway is linked to the emergence of planarian regeneration defects.

Although the details in the present manuscript are not precisely the same as in these studies and is focused on species of the cnidarian Hydra and on foot regeneration rather than head regeneration, the inspiration for this work and the framework for looking into what underlies differences regenerative potential in closely related species within a particular taxonomic group must surely come from these well-known papers which had a big impact when they were published. As such, I was very surprised to see that only one of these papers was cited (Sikes et al. 2013) and that it was only briefly mentioned in the Discussion.

The authors missed an opportunity to review these breakthrough studies of loss of regenerative ability in closely related species of flatworms and the implication of the Wnt/B-catenin pathway in modulating this ability. The overall approach taken in the present manuscript is also reminiscent of these studies, so I feel that they must be mentioned and briefly reviewed early in the paper, in the introduction, with a few clarifying statements about how studying this same general phenomenon in closely related species of Hydra in the context of foot regeneration centered on the same signaling pathway provides new insight. The way it is currently phrased in this manuscript's introduction, it sounds like there has been little progress in understanding of mechanisms driving loss of regenerative potential in any animal group, which is simply not

the case. Giving more context based on these studies and their findings will both enhance the present manuscript and give a better framework for focusing the current study.

This is the current text that needs to be substantially rewritten to reflect the suggestions above:

(lines 24-30)

“Many studies have already advanced our understanding of the genes and pathways that control regeneration in highly regenerative animals 3,4. However, we have little understanding of the mechanisms that drive loss of regenerative potential. Comparative approaches between closely related organisms with contrasting regeneration abilities offer a promising avenue to both reveal the mechanisms of regeneration, as well as uncover how regeneration has been shaped throughout evolution.”

In the Discussion, these same 2013 and 2023 planarian studies should be discussed again in the context of the results of the present study.

The current Discussion related to this is limited to a few sentences near the end of the Discussion and the focus is on differences in reproductive strategies between species in the *Oligactis* group of *Hydra* versus other groups:

(lines 503-511)

“One possibility is that the foot regeneration defect in *H. oligactis* evolved as a trade-off associated with semelparous reproduction. To explore this possibility, future research should explore the potential function of Wnt signaling in the induction of gametogenesis in *H. oligactis*. Similar trade-offs involving Wnt signaling may occur in other animals. For example, the semelparous planarian worm *Procotyla fluviatilis* cannot regenerate its head when cut near the tail due to excessive Wnt signaling activation upon injury; pharmacological downregulation of Wnt signaling rescues regeneration 44.”

This idea of different reproductive strategies having trade-offs with regenerative ability has been raised before in the literature and is also directly addressed in the Vila-Farré et al. 2023 paper (from the abstract):

“Our finding that Wnt signalling has multiple roles in the reproductive system of the model species *Schmidtea mediterranea* raises the possibility that a trade-off between egg-laying, asexual reproduction by fission/regeneration and Wnt signalling drives regenerative trait evolution. Although quantitative comparisons of Wnt signalling levels, yolk content and reproductive strategy across our species collection remained inconclusive, they revealed divergent Wnt signalling roles in the reproductive system of planarians.”

In summary, I suggest that the authors work to reframe and provide more context for this study by adding these papers to the introduction and the discussion to acknowledge that these ideas have been previously raised and explored, at least in a broader sense, and a similar approach was taken to identify what factors underline the differences in regenerative ability in a different group of animals and that this effort converged on identifying the Wnt/B-catenin pathway as key to these questions. Without this proper context, the reader is left with an incorrect impression of the novelty of the approach and the results from the present work.

We fully agree that it was an egregious oversight not to adequately discuss the planarian literature, and we have updated the manuscript accordingly. The details of these changes, which also highlight the novel aspects of our work, are provided in our response to Reviewer 2, who raised the same concern. In addition, we have expanded the discussion of reproductive trade-offs as suggested by this reviewer (577-586).

Beyond this inadequate framing and lack of proper context, the study itself is solid and provides a good basis and foundation for future study by setting up a new detailed comparison between two cnidarian species with differing abilities to regenerate their foot.

ROUND 1 REVIEWER 1 ATTACHMENT:

Campos et al. examine the genetic basis of why some hydra species fail to regenerate their foot region. They compared a species of *Hydra*, *Hydra oligactis*, that frequently does not regenerate their foot to regeneration of *Hydra vulgaris*. Through analysis of single cell sequencing of regenerating and non-regenerating *Hydra* species, they found that the *wnt* genes were some of the genes notably not being upregulated in *H. oligactis*. Remarkably, when the researchers stimulated *wnt* signaling with an agonist, they found the ability to regenerate the foot structures were restored. Conversely, when *wnt* signaling was inhibited in *H. vulgaris*, foot regeneration was delayed.

Regeneration is an interesting, but poorly understood process. It is especially unclear how and why some species regenerate robustly, and other species may fail to regenerate, or, in the case of *H. oligactis*, take longer. This approach of comparing *Hydra* species that exhibit different regeneration capacity allow some insight into how change in expression of early genes results in the loss of the ability to regenerate. This is exciting work that will be of interest to a large number of readers.

The manuscript is beautifully written, and the data are presented well. The methods are detailed and would allow for this approach to be used on other systems where there are closely related species that have different regenerative capacity. There is only one confusing part that could be clarified or presented differently. The graphs in Figure 3, showing the different levels of *wnt* expression in the two species have different axes on them, making the direct comparison of them confusing. Also, the double headed regenerates shown in Supplemental Figure 5 is spectacular! I don't know if this phenotype has been reported before in hydra before, but it seem that it would merit a paragraph in the discussion, at least.